



# Effect of ocean acidification and elevated temperature on growth of calcifying tubeworm shells (*Spirorbis spirorbis*): An *in-situ* benthocosm approach

Sha Ni[1,3], Isabelle Taubner[1], Florian Böhm[1], Vera Winde[2,4], and Michael E. Böttcher[2]

[1]GEOMAR, Helmholtz Center for Ocean Research Kiel, D-24148 Kiel, Germany
[2]Geochemistry & Isotope Biogeochemistry Group, Marine Geology Department, Leibniz Institute for Baltic Sea Research (IOW), D-18119 Rostock-Warnemünde, Germany
[3]Present address: Department of Geology, Lund University, 22362 Lund, Sweden
[4]Present address: LUBW, Institute for Lake Research, Langenargen, Germany

*Correspondence to:* Sha Ni (sha.ni@geol.lu.se), Isabelle Taubner (itaubner@geomar.de)

**Abstract.** The calcareous tubeworm *Spirorbis spirorbis* is a wide-spread serpulid species in the Baltic Sea, where it commonly grows as an epibiont on brown macroalgae (genus *Fucus*). It lives within a Mg-calcite shell and could be affected by ocean acidification and temperature rise induced by the predicted future atmospheric $CO_2$ increase. However, *Spirorbis* tubes grow in a chemically modified boundary layer around the algae, which may mitigate acidification. In order to investigate how increasing

5    temperature and rising $pCO_2$ may influence *S. spirorbis* shell growth we carried out four seasonal experiments in the "Kiel Outdoor Benthocosms" at elevated $pCO_2$ and temperature conditions. Compared to laboratory batch culture experiments the benthocosm approach provides a better representation of natural conditions for physical and biological ecosystem parameters, including seasonal variations. We find that growth rates of *S. spirorbis* are significantly controlled by ontogenetic and seasonal effects. The length of the newly grown tube is inversely related to the initial diameter of the shell. Our study showed no

10    significant difference of the growth rates between ambient atmospheric and elevated (1100 ppm) $pCO_2$ conditions. No influence of daily average $CaCO_3$ saturation state on the growth rates of *S. spirorbis* was observed. We found, however, net growth of the shells even in temporarily undersaturated bulk solutions, under conditions that concurrently favored selective shell surface dissolution. The results suggest an overall resistance of *S. spirorbis* growth to acidification levels predicted for the year 2100 in the Baltic Sea. In contrast, *S. spirorbis* did not survive at mean seasonal temperatures exceeding 24°C during the

15    summer experiments. In the autumn experiments at ambient $pCO_2$, the growth rates of juvenile *S. spirorbis* were higher under elevated temperature conditions. The results reveal that *S. spirorbis* may prefer moderately warmer conditions during their early life stages but will suffer from an excessive temperature increase and from increasing shell corrosion as a consequence of progressing ocean acidification.

*Copyright statement.* Authors 2017





## 1 Introduction

Atmospheric carbon dioxide ($CO_2$) is a primary substrate for life on Earth, but is also a major driver of global scale environmental change, causing ocean acidification (Greene et al., 2012), controlling climate variability (Retallack, 2002; Galeotti et al., 2016), and initiating mass extinctions (Jaraula et al., 2013; Veron et al., 2009). The recent rapid $CO_2$ rise from anthropogenic

emissions is a source of ocean acidification including pH reductions and alterations in fundamental chemical balances (Doney et al., 2007). Since the beginning of the industrial era, atmospheric $pCO_2$ rose from about 280 to 405 µatm (NOAA-ESRL, 2017, www.esrl.noaa.gov/gmd/ccgg/trends) due to human activities such as fossil fuel combustion, cement production and deforestation. At the same time surface seawater pH decreased by 0.1 units, corresponding to 30% increase in the hydrogen ion concentration (Raven et al., 2005; Cao and Caldeira, 2008). It is predicted to further decrease by 0.3 to 0.4 pH units until the

year 2100 when atmospheric $pCO_2$ levels may reach 950 µatm (IPCC, 2013). By the end of this century, the average surface ocean pH could be lower than it has been for more than 50 million years (Caldeira and Wickett, 2003) with severe consequences for marine calcifying organisms (Orr et al., 2005; Andersson et al., 2008; Erez et al., 2011).

The $CO_2$ rise also caused an increase of sea surface temperatures (SST) of about 1°C on a global scale (European Environment Agency, 2015). However, mid and high latitude SSTs are more variable and increase more rapidly than the global

average. For instance, the Baltic Sea annual mean SST warmed by up to 1°C per decade between 1990 and 2008 (Elken et al., 2015). Warming of up to 6°C and prolonged summer heat waves are expected until the end of 21st century (HELCOM, 2007; Gräwe et al., 2013). Rising temperatures and summer heat waves may increasingly affect mid/high-latitude marine ecosystems in the future, e.g., through micro-/macroalgae ecological functions, impacts on food-web structures or reduced reproduction (Knight-Jones et al., 1972; Graiff et al., 2015a; Werner et al., 2016). Stress from elevated temperatures can cause a depletion

of organism's energy supplies resulting in energy deficiencies and increased mortality (Ivanina et al., 2013).

Coastal water $pCO_2$ and pH can be much more variable than that of the open ocean due to the effects of run-off, upwelling, eutrophication, atmospheric deposition and remineralisation (Doney et al., 2007). The Baltic Sea is an intra-continental non-tidal brackish water environment with highly variable seasonal dynamics of $pCO_2$ and pH. Annual pH ranges vary from 8.1-8.4 in the Kattegat area to 7.4-8.4 in the less saline eastern Baltic (Havenhand, 2012). Kiel Fjord and Eckernförde Bay are narrow

coastal embayments in the western Baltic Sea. Surface water data from Kiel Fjord show a seasonal pH range from 7.3 to 8.5 (NBS scale) with $pCO_2$ varying from 385 to 2500 µatm (Thomsen et al., 2010, 2013; Wahl et al., 2015). Significant variations in pH and $pCO_2$ were observed along the coast line of the Kiel bight (Winde et al., 2017). In the *Fucus* meadows of Eckernförde Bay diurnal pH variations from 7.3 to 7.8 were found during an upwelling episode, while during normal summer conditions pH varied between 8.0 and 8.4 (Saderne et al., 2013). These observed ranges of $pCO_2$ and pH by far exceed the predicted levels

at the end of the 21$^{st}$ century. Therefore, the question arose: Are calcifying organisms living under such dynamic conditions better adapted for future ocean acidification?

Consequences of ocean warming and acidification for marine organisms have been investigated in many studies (e.g., Reynaud et al., 2003; Marshall and Clode, 2004; Veron et al., 2009; Saderne and Wahl, 2013; Wisshak et al., 2013; Cornwall et al., 2016; Wahl et al., 2016). However, only few studies investigated combined effects of simultaneously increased temperature



and $CO_2$ on whole ecosystems (Wahl et al., 2015). To study the combined impact of temperature rise and elevated $CO_2$ on typical marine calcifiers from the Baltic Sea, we carried out experiments in the Kiel Outdoor Benthocosms ("KOB", Wahl et al., 2015) to investigate calcification of the serpulid tubeworm *Spirorbis spirorbis* under near-natural habitat conditions as sessile epibionts on the thalli of *Fucus* seaweeds.

The brown algae *Fucus vesiculosus* and *Fucus serratus* are among the most widespread brown seaweed found on the coasts of the Baltic Sea. The pH in the seaweed ecosystems shows significant diurnal variations due to photosynthesis (high pH during the day) and respiration (low pH during the night) (Saderne et al., 2013). A diffusive boundary layer (DBL) of typically 50 µm to 2 mm thickness surrounds the algal thalli depending primarily on the flow conditions (Larkum et al., 2003; Spilling et al., 2010; Hurd and Pilditch, 2011; Wahl et al., 2016). Micro- or macroepibionts living in the DBL are affected by conditions

with variable concentrations of chemical compounds (e.g. $O_2$, DIC and pH) that are created by algal bioprocesses (Larkum et al., 2003). In the DBL of *F. vesiculosus* pH was found to increase by up to 1.5 units from dark conditions to bright daylight (Spilling et al., 2010; Wahl et al., 2016). Consequently, this surface boundary layer of the algae can potentially provide a shelter from ocean acidification during daylight time (Hendriks et al., 2014; Pettit et al., 2015).

    Water temperature significantly influences growth, photosynthesis and metabolism of algae. Optimal temperature for growth

of Baltic *F. vesiculosus* is in the range of 15 to 20°C, but growth decreases rapidly when the water temperature exceeds 27°C for several days (Graiff et al., 2015a). High temperatures may therefore have indirect adverse effects on epibionts, like *S. spirorbis*, because the ecological functions of their host algae may be reduced or damaged.

    *Spirorbis spirorbis* (Linnaeus, 1758) is a millimeter-sized, coiled calcareous tubeworm which belongs to the family Serpulidae, subfamily Spirorbinae (class Polychaeta). The Spirorbinae originated in the later Mesozoic and became common during

the latest Cretaceous (Ippolitov and Rzhavsky, 2014). The tube of *S. spirorbis* is sinistral, planospiral, unsculptured, commonly with a small, peripheral flange increasing the area attaching to the substrate (Fig. 1; Ippolitov and Rzhavsky, 2015). *S. spirorbis* usually lives attached to seaweeds and eel grass in shallow sublitoral and intertidal marine environments (Ippolitov and Rzhavsky, 2015). It favours toothed wrack (*Fucus serratus*), bladder wrack (*Fucus vesiculosus*, Fig. 1) and kelp (*Laminaria* spp.), and rarely grows on other substrates like rocks or other algae (De Silva, 1962; O'Connor and Lamont, 1978; Qian, 1999).

It is a common species of the Baltic Sea, where it lives in coastal macrophyte meadows characterised by large pH variations (>1 pH unit) and frequent aragonite under-saturation ($\Omega_{arag}$>0.6, Saderne et al., 2013).

    *S. spirorbis* shells are purely calcitic. No or only questionable indications of aragonite have been reported (Ippolitov and Rzhavsky, 2015). The tubes consist of Mg-calcite with about 10 mol% $MgCO_3$ (Bornhold and Milliman, 1973) (Ni et al., in prep.), which has a similar solubility as aragonite (Plummer and Mackenzie, 1974; Walter and Morse, 1984; Morse and

Mackenzie, 1990). Obviously, *S. spirorbis* is able to prosper in temporarily $CaCO_3$ under-saturated water. Other serpulid worms have even been reported to calcify in abyssal waters below the calcium carbonate compensation depth (Kupriyanova et al., 2014).

    Previous work on Baltic *S. spirorbis* in laboratory experiments (Saderne and Wahl, 2013) found significantly reduced growth only at pH values lower than 7.7 ($\Omega_{arag}$<0.8). The study confirmed that the tubeworms were able to calcify in aragonite under-





saturated water ($\Omega_{arag}$<1). This points to a high short-term tolerance for ocean acidification for at least some of the serpulid worm species.

Several recent ocean acidification experiments have included serpulid worms of a variety of species. Most studies focused on the tropical species *Hydroides elegans* (Lane et al., 2013; Chan et al., 2012, 2013; Mukherjee et al., 2013; Li et al., 2014). The

results indicated reduced growth, increased porosity and reduced mechanical strength of the worm tubes, as well as increased mortality of larvae at lowered pH (<7.9).

Field experiments in subtropical settings show reduced serpulid population counts at lowered pH (Campbell and Fourqurean, 2014; Peck et al., 2015). In a Mediterranean sea-grass meadow, naturally acidified by volcanic $CO_2$ seeps, calcareous serpulids were absent at sites with high pCO ($pH=7.1$; $\Omega_{arag}=0.6$) (Cigliano et al., 2010). In this area the specialized tubeworm species

*Simplaria* sp. dominates serpulid populations in intermediate-pH habitats (pH $\sim$7.4, $\Omega$ $\sim$1.1) (Lucey et al., 2016). Experiments with *Hydroides crucigera* in a temperate setting, on the other hand, showed only moderate impacts of acidification on serpulids even in undersaturated water ($\Omega_{arag}$<0.8), including a shift in tube mineralogy (Ries et al., 2009; Ries, 2011).

In the present study, we compare growth rates and corrosion features of *Spirorbis spirorbis* grown under ambient and elevated pCO$_2$ and temperature conditions in four seasonal experiments to test their sensitivity to ocean acidification and warming. Our

results also provide new information about the life cycle and shell microstructure of *S. spirorbis*. The growth experiments were carried out in the Kiel Outdoor Benthocosms under near natural conditions, exposed to the weather and water conditions of the Kiel Fjord, by using a flow-through setup with water pumped directly from the fjord (Wahl et al., 2015).

## 2 Material and methods

### 2.1 Sampling

Healthy *F. vesiculosus* plants bearing intermediate amounts of live *S. spirorbis* were collected for 4 seasonal experiments in less than 1.5 m water depth in Eckernförde Bay (54°27'N, 9°53'E, Western Baltic Sea, Germany) in March 2013, June 2013, October 2013 and January 2014. The location is described in detail by Saderne et al. (2013) and Winde et al. (2017). Every individual plant contained nearly the same volume of blades with a similar amount of *S. spirorbis* tubes attached. The collected plants were transported in a cool-box to GEOMAR (Kiel, Germany) for subsequent treatments.

### 2.2 Culturing

The samples were stained outdoor at the quay at GEOMAR in a closed transparent plastic box, for three days in Kiel Fjord seawater with $\sim$50 mg/L calcein bubbled with ambient air. *S. spirorbis* were fed at the start of the staining with *Rhodomonas* algae. The staining box was placed in a flow through water trough with seawater pumped from the Kiel Fjord to keep the temperature close to ambient conditions in the Fjord. The absorption of the dye into newly grown tubes provides a well defined

starting point for growth under the experimental conditions. After three days staining, twelve individual *Fucus* plants were




transplanted into the twelve subunits of the Kiel Outdoor Benthocosms (Wahl et al., 2015), fixed on a plastic grid at the bottom of the basins under 0.4 m of water. The incubations started immediately after staining.

The twelve benthocosm subunits were assigned to four treatments. Each treatment had three replicates (Wahl et al., 2015): "control treatment" with ambient $pCO_2$ (380 - 400 µatm) and water temperature, "+$CO_2$ treatment" with 1100 µatm $pCO_2$ in

the headspace of the subunit, "+T treatment" with water temperature elevated by 5°C over ambient conditions, and "+$CO_2$+T treatment" as a combination of both elevated $pCO_2$ and temperature. These conditions are considered as representative for acidification and temperature changes at the end of 21$^{st}$ century (Wahl et al., 2015). Each benthocosm subunit had a volume of 1500 L and was continuously flushed with ambient fjord water, pumped from 1 m below the surface at a flow rate of about 65 L/h. Water in the subunits was additionally mixed by artificial waves with a frequency of 30 waves per hour. Four seasonal

experiments were carried out: "spring" (04 April - 19 June 2013), "summer" (4 July - 17 September 2013), "autumn" (10 October - 17 December 2013), "winter" (16 January - 1 April 2014). In total each subunit contained 21 *Fucus* plants, but only one with *Spirorbis*, and a fauna of mollusks, arthropods and echinoderms. Details of the KOB setup and experimental parameters are described in Wahl et al. (2015), Graiff et al. (2015b) and Werner et al. (2016). After 10 - 11 weeks of incubation, the twelve algal plants with *S. spirorbis* were collected from the benthocosms for freeze drying and further analyses.

## 2.3 Measurements and statistics of *S. spirorbis* growth

*S. spirorbis* specimens were peeled off from the algal surfaces and photographed under an epifluorescence microscope. The initial and final diameter (in millimeter) of *S. spirorbis* shells and the length of the newly grown tube segments (mm) were measured after observing the position of the staining line (Fig. 2). The absolute tube length increase (mm) was measured as the length of the newly formed external arc of the tube between the staining front and the terminal tube edge, following Saderne

and Wahl (2013).

From the spring, summer and and autumn experiments *S. spirorbis* tubes were collected from some basins for chemical analysis (Ni et al., in prep.). From each basin the newly grown tube parts of up to 20 specimens were cut off at the stain line, pooled, bleached, washed, dried and weighed. Bleaching was carried out using sodium hypochlorite with 1% active chlorine.

The measured length increase and final diameters were normalized by the initial diameter. In our analysis we compared the

resulting five growth parameters: (1) Initial diameter, $D_i$, (2) final diameter, $D_f$, (3) growth, $Gr$, (4) growth/initial diameter, $Gr/D_i$, and (5) final diameter/initial diameter, $D_f/D_i$. In order to test the robustness of the different parameters we measured $Gr$ and $D_f$ of specimens with similar initial diameters in the autumn, winter, and spring populations. The results showed that $Gr$ measurements were more sensitive in detecting growth differences than the $D_f$ measurements.

Normalization to the initial diameter was applied because growth of *S. spirorbis* tubes is strongly size-dependent. However,

as the dependence is not strictly linear (see Section 3.5) we based all growth rate comparisons on the condition that the initial diameters of the starting populations were in the same range. The clearly bi-modal populations in the autumn experiment were treated separately (autumn-big and autumn-small). The summer populations and autumn-small populations, which both were dominated by juveniles, differed significantly from the autumn-big, winter and spring populations, dominated by adults. Therefore no comparisons were carried out between these two sets of populations, because there was very little overlap in





the initial sizes (compare Results Section, Fig. 5). Initial diameters of the summer and autumn-small populations overlapped to a high degree, but the medians differed significantly. In this case we selected sub-populations with a homogeneous initial diameter range, so that the median initial sizes were similar. For the autumn-big, winter and spring populations the initial diameters were not significantly different, as verified by Tukey's HSD tests.

Three-, two-, one-way ANOVA and Tukey's HSD tests were used for testing statistical significance of differences between the median values from different treatments and seasons. Each treatment had three replicates. Median values were calculated for each replicate based on the measured values from each basin. In the three-way ANOVA, the three factors were temperature, $pCO_2$ and season. The temperature and $pCO_2$ factors had two levels, elevated and ambient. It should be kept in mind that the season factor here is a multiple factor which includes a range of parameters/conditions such as temperature, pH, saturation

state, nutrients and ontogenetic effects of *S. spirorbis*. Assumption of normality of the models' residuals and homogeneity of residual variances were tested with Shapiro-Wilk's tests and box plots, respectively. Statistical analyses were conducted with R (Version 3.2, cran.r-project.org), PAST (Version 3.13, Hammer et al., 2001) and Microsoft Excel (Data Analysis Tool). A probability value of <0.05 was considered significant.

## 2.4 Microstructures

For localization of the calcein stain line *S. spirorbis* specimens were photographed with an epifluorescence microscope (AxioScope A1, Carl Zeiss, Germany). Polished longitudinal and cross sections were used for electron microscopy. Backscatter electron images (BEI) and element concentration maps of calcium were taken with a JEOL JXA 8200 "Superprobe" electron microprobe (EMP) at GEOMAR Kiel, Germany. High resolution (2-3 μm per pixel) maps of calcium were recorded with 50 nA beam intensity at 15 kV, eight accumulations and 100 ms dwell time. Internal structures of stained skeletons were imaged

on polished cross sections with a Zeiss Axio Imager.M2 microscope using white field and differential interference contrast (DIC). The Cy3 filter set was applied for detection of calcein.

## 2.5 Seawater Chemistry

Temperature and $pH_{NBS}$ in all benthocosm treatments and the fjord water inflow were logged at two-hour intervals by GHL temperature sensors (PT1000) and pH glass electrodes, respectively. Air $pCO_2$ in the head space of the $+CO_2$ treatment subunits

was monitored using infrared spectroscopy and kept at a constant level as described by Wahl et al. (2015).

Additionally, $pH_{NBS}$ values were measured daily using a Seven Multi1InLab Expert Pro (pH, Mettler Toledo GmbH, Giessen, Germany). The pH electrode was calibrated with NBS pH-buffer (4.001, 6.865) (Winde et al., 2014, 2017). Discrete water samples were taken as described by Wahl et al. (2015) and analysed for total alkalinity (TA) two times a week, as well as for dissolved inorganic carbon (DIC) on a monthly base. Water samples for DIC analysis were filled bubble-free into 50

mL Winkler bottles, poisoned by the addition of one drop of saturated mercury chloride ($HgCl_2$) solution and measured via coulometric titration (Johnson et al., 1993). TA samples were filtered through 0.45 μm Minisart syringe filters (Sartorius SFCA, Sartorius) and measured by potentiometric titration using 0.01 M HCl (with added appropriate amounts of NaCl) with a Schott



titri plus and an IOline electrode A157. The titration cell was kept at 25°C. Measurements were calibrated using certified seawater standards (Dickson et al., 2003, 2007).

The speciation in the dissolved carbonate system, including the carbonate ion concentration, was calculated from $pH_{NBS}$ and TA using the code of the CO2SYS software package for MATLAB, Version 1.1 (Lewis and Wallace, 1998; van Heuven et al.,

2011) with constants recommended for best practice (Dickson et al., 2007; Orr et al., 2015), i.e. $K_1$ and $K_2$ from Lueker et al. (2000), $K_S$ and $K_B$ from Dickson (1990), $K_F$ from Dickson and Riley (1979), $K_W$, $K_{1-3P}$ and $K_{Si}$ from Millero (1995) and the total boron-salinity relationship from Uppström (1974). The $K_1$ and $K_2$ constants from Lueker et al. (2000) are defined for a salinity range from 19 to 43, while the brackish Kiel Fjord water ranged from about 10 to 20 psu during the experiments. An alternative set of equations for $K_1$ and $K_2$ is available from Millero (2010) for salinities as low as 1. However, as discussed in

Orr et al. (2015) applications of the latter showed discrepancies on different pH scales. Therefore, we used the Lueker et al. (2000) constants. Using the latter to calculate carbonate ion concentrations at salinities as low as 10 psu resulted in offsets of less than 0.5 % compared to the Millero (2010) constants, which is negligible for our interpretations.

Salinity and concentrations of Ca, Si and P were measured in all benthocosm treatments two times a week. Si and P concentrations were usually too low to have a substantial impact on alkalinity (Winde et al., in prep.). Alkalinity and salinity behaved

conservatively in our experiments and showed no significant systematic variability on diurnal timescales (Winde et al., in prep.; Wahl et al., 2015). Calcium concentrations ranged from about 3.5 to 6 mM and were closely coupled to salinity ($R^2$>0.9).

The saturation state in the benthocosm treatments with respect to the calcium carbonate of *S. spirorbis* tubes was calculated considering the shell composition. It has been shown, that the thermodynamic stability of biogenic Mg-calcites differs from pure calcite (Plummer and Mackenzie, 1974; Busenberg and Plummer, 1989) and varies with the Mg content. The $MgCO_3$

content of *S. spirorbis* tubes is about 10±1 mole% (Ni et al., in prep., Bornhold and Milliman, 1973). Unfortunately, the solubility of *S. spirorbis* was not explicitly determined, so far. According to different experimental studies biogenic Mg-calcite with about 10 mol% $MgCO_3$ has a solubility which is in thermodynamic equivalence to aragonite (Walter and Morse, 1984; Morse and Mackenzie, 1990; Andersson et al., 2008). Therefore, the saturation state with respect to aragonite ($\Omega$) was taken as an estimate for the Mg-calcite forming the *S. spirorbis* shell. It should, however, be kept in mind that the solubility of biogenic

Mg-calcites may not only differ with shell composition, but may also depend on crystal ordering, trace element impurities and other mineralogical factors (Mackenzie et al., 1983). Most of these factors increase the solubility of Mg-calcite.

Saturation states in the benthocosms at the measured *in-situ* temperatures and salinities were calculated from carbonate ion concentrations, calcium ion concentrations and the apparent solubility constant ($K^*_{sp}$) of aragonite (Mucci, 1983):

$$\Omega = [Ca^{2+}] \cdot [CO_3^{2-}]/K^*_{sp} \tag{1}$$

Only pH and temperature were measured with two-hourly resolution (Wahl et al., 2015). All other parameters ($[Ca^{2+}]$, TA, salinity, Si, P) were interpolated to calculate diurnal variations of $\Omega$ (Fig. S1). Linear interpolation is justified by the conservative behaviour of these properties. The resulting two-hourly resolved time series of $\Omega$ were used to estimate the mean saturation state and the percentage of time when treatments were undersaturated with respect to *S. spirorbis* tube calcite.



Average diurnal amplitudes of saturation state, pH and temperature were calculated as follows: The pH and temperature time-series from Wahl et al. (2015) and the resulting saturation values have an interpolated resolution of 10 minutes. We averaged all values of the period of interest into 24 one-hour bins, resulting in a mean value for each hour of the day. The minimum and maximum values of the resulting mean diurnal cycle define the mean diurnal amplitude. The resulting values

were averaged for each of the four different treatments. Each experimental period was subdivided into four sub-periods with durations of 17-19 days and mean diurnal amplitudes of each sub-period were calculated as explained above.

We calculated a simplified insolation index for all sub-periods to compare with the diurnal pH variations. For this we averaged the maximum daily irradiance values available from the GEOMAR meteorological observatory[1], situated close to the benthocosms. Mean maximum irradiance varied between ∼90 W/m$^2$ in December and ∼900 W/m$^2$ in June. The resulting

values were multiplied with the fractional daylight period (0.31 in December to 0.71 in June), defined by sunrise and sunset. The values are expressed as percentages of maximum insolation, ranging from 0.04 in December to 1.0 in June.

## 3   Results

### 3.1   Seawater carbonate chemistry and saturation state

The calcium carbonate saturation state of the seawater ($\Omega$) in all basins was dominantly controlled by the pH. Average diurnal

cycles showed a minimum in pH and $\Omega$ around sunrise followed by a late afternoon maximum (Fig. S2). The pH values showed strong diurnal fluctuations in all treatments. Average day/night pH differences were smallest (<0.05) in December 2013 and largest (up to 0.6) in June, July and August 2013, as well as in February/March 2014 (Fig. 3). The pH amplitudes showed a clear correlation to insolation at low light levels (<20% of maximum insolation, $R^2$=0.72, n=20, p<0.0001). At higher light levels only the $CO_2$-enriched treatments showed a weak light dependence (high $CO_2$: $R^2$=0.36, n=22, p=0.003; ambient $CO_2$:

$R^2$=0.02, n=22, p=0.5; Fig. 4). Generally, pH values declined from the spring to the autumn experiment and reached a minimum in November and December (Fig. 3).

Saturation states closely followed the pH dynamics. Average saturation was highest during the spring experiment when all treatments were generally oversaturated with respect to aragonite and Mg-calcite ($\Omega$>1). Basin waters were undersaturated ($\Omega$<1) only during 6 to 51% of the time during the spring experiment (Table 1). The lowest saturation states occurred during

the autumn experiment with $\Omega$<1 during 81 to 100% of the experiment duration. Average autumn saturation ranged from 0.6 to 0.8 (Table 1). It was only slightly elevated during daytime ($\Omega_{max}$ of 0.6 to 1.1, Fig. 3). Average day-night differences in $\Omega$ were largely tracking the diurnal pH amplitudes with smallest differences during the autumn experiment (<0.1 in December 2013) and large fluctuations in February/March, June and July (up to 2.2, Fig. 3).

---

[1]www.geomar.de/service/wetter





### 3.2 *Spirorbis spirorbis* tube size and ontogenetic cycle

The sizes that the *S. spirorbis* shells reached before the experiments in their natural environment are indicated by the initial diameters. They reflect the size distributions under natural conditions. In contrast, the final diameters of our specimens reflect changes from the initial sizes under experimental conditions. Note that only stained specimens were included in the analysis.

Therefore, juveniles that settled during the experiments and specimens that did not calcify during the staining were not included.

The final and initial diameters of 2782 stained and photographed *S. spirorbis* tubes from all four seasonal experiments were in a range of 0.2 mm to 4.0 mm (Fig. 5). The tube with the biggest final diameter (∼4.0 mm) was found in the winter experiment. The smallest measured shell diameters (0.2 mm) occurred in summer and autumn (Fig. 5 B,C). The size distributions of the shells indicate distinct populations that, in summer and autumn, were separated by a minimum in shell counts at a diameter

of about 1.3 mm (Fig. 5 B,C). Accordingly we classified *S. spirorbis* specimens into two general populations: "juveniles" (diameter <1.3 mm) and "adults" (diameter>1.3 mm). Seed et al. (1981) observed reproduction of *S. spirorbis* at a shell diameters >1.9 mm. Therefore, our "adult" populations may include immature specimens.

"Adult" specimens were observed in the starting populations of all seasons (Fig. 5 B-E), including spring and summer, which is in accordance with the maximum life span of *S. spirorbis* of about 1.5 years (Seed et al., 1981). Most juvenile specimens grew

to "adult" sizes during the ∼10 weeks duration of the experiments (Fig. 5 F-I). The majority of *S. spirorbis* in the maximum size range (>3 mm) occurred by the end of the winter and spring experiments (March, June, Fig. 5 A,H,I). The initial shell diameters of the *S. spirorbis* autumn population showed a clear bi-modal distribution (Fig. 5 C). A juvenile population with a modal diameter of 0.6 mm ("autumn-small") was clearly separated from an "adolescent/adult" population with a modal diameter of 1.8 mm ("autumn-big"). Initial diameters in the intermediate range of 1.4 mm - 1.5 mm were scarce. A similar size

distribution was found in the summer experiment. However, the "adult" population had very few specimens in summer (Fig. 5 B).

Juveniles occurred in all four seasons but were rarely observed in winter and spring. The proportion of juveniles in the initial populations decreased from July (Fig. 5 B) to April (Fig. 5 E). Accordingly, the majority of the *S. spirorbis* specimens at the start of the summer and autumn experiments were in the juvenile stage (<1.3 mm), while the winter and spring experiments

were dominated by "adults" (Fig. 5). The modal initial diameter increased systematically with the sequence of the seasons from July (∼0.7 mm, Fig. 5 A,B) until April (∼2.4 mm, Fig. 5 A,E). The spring, winter and autumn-big populations started with similar initial diameters (modes of 1.8 to 2.5 mm, Fig. 5 C-E) and all grew into a typical final diameter range (modes of 2.5 to 2.8 mm, Fig. 5 G-I) representing the most common size of adult *S. spirorbis*.

As shown above the diameter increase of *S. spirorbis* tubes during the experiments strongly depended on the season and

the initial size distribution of the populations. Diameter increases ranged from 4 µm/day for the adult-dominated population in spring to 20 µm/day for the juvenile population in autumn. Modal diameter increases of the summer, autumn-big and winter populations were similar (Fig. 5) with values of about 10 µm/day. This ontogenetic influence has to be taken into account when interpreting growth rates in terms of temperature and saturation state.



### 3.3 Tube microstructure

SEM pictures of *S. spirorbis* sections (Fig. 6) show a relatively rough and irregular outer tube wall surface whereas the inner surface is smooth. The internal wall structures consist of convex-forward lamellae or chevrons (Fig. 6b). New lamellae were laid down by the worm on the anterior tube surface, forming curved convex-forward layers, wrapping the end of the tube wall to completely cover the end of the anterior tube wall with a new layer. Thin crescent pores exist in the wall interior between the chevrons (Fig. 6b). These pores taper towards the inner and outer rims of the tube wall where the chevron lamellae fuse into a dense, calcium-rich wall (Fig. 6). The high calcium concentrations indicate that not organic but strongly calcified dense layers armour the inner and outer tube wall surfaces.

A comparison of a cross section (Fig. 6a) and a longitudinal section (Fig. 6d) through *S. spirorbis* shells reveals the complex shape of the growth lamellae. The convex-forward layers are additionally curved upward, forming convex-upward lamellae in longitudinal sections. The convex-upward lamellae were built upward successively from the bottom on both sides of the tube and then converge at the tube top. The growth direction is indicated by the convex layering.

In addition, the inner and outer sides of each convex-forward layer of the tube walls are asymmetric (Fig. 7). The fluorescent, stained skeleton outlines the pattern of lamellae which were accreted during the 3 days staining period. The lamellae cover a large area of the inner wall surface, but hardly grow over the outer wall surface.

The bottom of the tube, which was attached to the substrate, is relatively thin and characterized by parallel planar lamellae. An idealized sketch of the *S. spirorbis* tube structures is shown in Fig. 8. Where the wall of a new whorl attached to an older whorl it formed a thickened wedge-like structure partly filling the gap between the old and new whorl (Fig. 6d, 8). These wedges are usually calcium-rich, densely calcified (Fig. 6e), increasing the stability of the shell. The tube diameter of the whorls and the tube wall thickness generally increased as the *S. spirorbis* shell grew (Fig. 6d). The wall thickness ranges from about 30 to 180 μm. It is thicker in the fully developed shell parts and tapers towards the tube mouth (Fig. 6a, b).

### 3.4 Shell corrosion

Shell corrosion (Fig. 9) occurred in all treatments during all seasons, but was most commonly observed in the high $pCO_2$ treatments of the autumn and winter experiments (Fig. 10, Table 2). In the basins of these treatments up to 75 % of the specimens showed corroded shells. On average, the proportion of corroded samples ($P_{corr}$) was highest in the autumn $+CO_2$ treatment and in the winter $+CO_2+T$ treatments, with treatment averages of 58% and 62%, respectively. In contrast, corrosion was nearly absent in the control treatments, where in all four seasons $P_{corr}$ values were lower than 1.5%. Additionally, corroded specimens were nearly absent in all spring treatments, except for the $+CO_2+T$ treatment.

The percentage of corroded samples was clearly related to the saturation state (Fig. 10). Except for one basin from the spring $+CO_2+T$ experiment $P_{corr}$ was below 10% when average saturation ($\Omega$) was above 1. For average saturation $\Omega>2$ corroded shells were completely absent. On the other hand, although $P_{corr} = 0\%$ was observed in basins with an average saturation as low as 0.8 (basin D2, autumn control), corrosion frequencies generally increased in undersaturated basins. For $\Omega<1$ we observed a





significant inverse correlation between $P_{corr}$ and saturation state:

$$P_{corr}(\%) = -143\pm72 \cdot \Omega + 131\pm51; R^2 = 0.54; n = 17; p < 0.0008 \qquad (2)$$

Notably, shells grew significantly even in undersaturated waters. Thus corrosion selectively affected the previously grown parts of the shell (Fig. 9b).

$P_{corr}$ was independent of temperature in autumn, winter and spring ($R^2=0.03$, n=42, p=0.28), but temperature may have fostered corrosion and bioerosion in the summer experiments. Ambient temperature treatments of the summer experiments showed very low $P_{corr}$ values (Fig. 10). However, the few recovered samples from the elevated temperature experiments were highly corroded and showed very little net growth. Unfortunately, because very few specimens were recovered from these treatments of the summer experiment, $P_{corr}$ values could not be determined.

Strong bioerosion by microborers was observed in a cross section of a summer control specimen (Fig. 11). Numerous microborings of about 5 to 45 μm diameter affected the outer tube wall. The microborings penetrated the whole tube wall. This is in contrast to the shell corrosion of the other seasons, which mostly affected the outermost layer of the tube wall (Fig. 9b).

### 3.5    Growth rate

The length of new tube segments that grew during an experiment (Fig. 2: "growth", Gr) varied considerably between popula-
tions and seasons, ranging from less than 0.1 mm up to 7.3 mm. This corresponds to a range in growth rates of 1 to 100 μm/day. The longest newly grown tube in all experiments (7.3 mm) occurred in the autumn-small population. Growth was found to be inversely correlated with the initial diameter of the shells ($D_i$), i.e. smaller tubeworms generally grew faster than bigger ones (Fig. 12). The correlation is highly significant

$$Gr(mm) = -1.1\pm0.05 \cdot D_i(mm) + 5.17\pm0.09; R^2 = 0.41; n = 2783; p = 0; \qquad (3)$$

for $D_i$ ranging from 0.2 to 3.5 mm.

    Growth of the winter populations showed the highest variability of all treatments, ranging from 0.4 mm to 6.3 mm (Fig. 12). Growth rates and initial sizes in winter were similar to those of the autumn-big populations. This indicates that the tubeworms from these two experiments were in the same developing stage, although they represented different generations of *S. spirorbis* populations (Section 3.2, Fig. 5).

In a subset of specimens from the spring and autumn (control, +CO$_2$+T) and the summer (control,+CO$_2$) experiments average weights of newly grown tube segments, $W_t$, were determined (Table S1). The results show similar weight increases in spring and summer of 0.1 - 0.9 and 0.2 - 0.6 mg/shell, respectively. In contrast, $W_t$ values in autumn were significantly larger, ranging from 1.2 to 2.1 mg/shell. As shown above Gr varied seasonally (Fig. 12). For the weighed specimens mean Gr ranged from 2.2 to 3.8 mm and 3.8 to 5.4 mm in summer and autumn, respectively. It was only 1.0 to 2.3 mm in spring. We accordingly
normalized $W_t$ by Gr. This resulted in overlapping $W_t$/Gr ranges for spring and autumn of 0.1 - 0.4 and 0.3 - 0.4 mg/mm of tube, respectively (Fig. 13). The summer shells increased their weights by only 0.1 - 0.2 mg/mm of tube.

    Generally, this is in agreement with smaller final shell sizes in summer (Fig. 5) and consequently smaller tube widths (Fig. 2). Assuming a cylindrical tube geometry and a constant wall thickness of 0.1 mm the measured tube width values (Table 3) allow





to estimate average shell densities of $1.1 \pm 0.3$ and $1.8 \pm 0.3$ g/cm³ ($\pm 1$sd) for the summer and autumn specimens, respectively. This indicates that density and/or tube wall thickness of the summer tubes was $38 \pm 13\%$ lower compared to the autumn tubes. The difference is significant (t-test, p=0.005).

### 3.5.1 Treatment effects

Only very few broken and strongly damaged *S. spirorbis* specimens could be recovered from the elevated temperature treatments ($+T$, $+CO_2+T$) of the summer experiment. In these experiments a temperature-driven collapse of the grazer community had caused epiphytic overgrowth of *Fucus* thalli and *S. spirorbis* tubes leading to an increased mortality (Werner et al., 2016). Except for these elevated temperature summer treatments there was no significant treatment influence ($pCO_2$ or T) on growth in spring, summer, winter or autumn (big population). Notably, elevated $pCO_2$ had no detectable influence on growth in any of

the four seasonal experiments (Fig. 14).

    In the autumn-small population, temperature caused a significant increase of growth, but only under ambient $pCO_2$ condition (Fig. 14; three-way ANOVA and Tukey's HSD tests, p=0.03 for $Gr/D_i$, p=0.04 for $D_f/D_i$). There were marginally significant interactions ($p<0.1$) among the factors temperature, $pCO_2$ and season. Each factor influenced the growth parameters in each experiment differently due to the effects of the other two factors.

No significant correlation between saturation ($\Omega$) and growth parameters ($Gr$, $Gr/D_i$, $D_f/D_i$) was found ($p>0.14$ to 1.0, $R^2=$ 0.00 to 0.40, seasonal basin data in Tables 1 and S2). The extension rates of *S. spirorbis* tubes were not negatively impacted by the saturation state of seawater. In contrast, weight increase ($W_t/Gr$, Table S1, Fig. 13) showed a significant positive correlation with saturation state in spring ($R^2=0.94$, n=6, p=0.002) and autumn ($R^2=0.68$, n=6, p=0.04). Weight increases of the autumn specimens were similar to spring, although the tubes formed in undersaturated water. In the summer experiment no significant

correlation was observed ($R^2=0.48$, n=6, p=0.13), but the data lie close to the spring trend line ($R^2=0.88$, $p<10^{-5}$ for summer and spring combined, Fig. 13).

### 3.5.2 Seasonal effects

As described in Section 3.2 we observed a significant seasonal variation in the proportion of juvenile specimens (Fig. 5), indicating limited reproductive activity during the cold seasons. In spring and winter less than 10% of the stained specimens

were juveniles ($D_i<1.3$ mm) while there were more than 84% juveniles in summer and autumn. As a consequence $D_i$ values were seasonally biased, which can explain at least some of the seasonal variations of $Gr$ (Fig. 12).

    In order to detect additional seasonal impacts on *S. spirorbis* tube growth we compared the juvenile populations in the control treatments of the summer and autumn experiments. Populations with similar mean $D_i$ were selected (Table 3). No significant seasonal impact on growth ($Gr$) was found (Fig. 15). However, the final diameters of the autumn-small population

were significantly larger than those of the summer experiment (Tukey's HSD test, $p<0.01$, Fig. 5F-G). Additionally, the width of the newly grown tubes (Fig. 2) differed significantly between the two seasons (Tukey's HSD test, $p<0.001$). The tubes that formed in autumn were wider than the summer tubes. Two-way ANOVA of tube width values from the control and $+CO_2$ treatments of the two seasons (Table 3) indicated no treatment effect (p=0.21) but a significant seasonal impact ($p<0.0001$).



There was no significant difference in growth of the adult populations between the winter and autumn experiments (Fig. 12). There was no influence of temperature, $pCO_2$ or season on $Gr/D_i$ of these populations (three-way ANOVA, p>0.67). All populations that had "adult" sizes at the start of the experiments (spring, autumn-big, winter) grew to a similar final size distribution at the end of the experiments (Fig. 5 G-I). Consequently, because the initial diameters of the winter and autumn-big populations were generally smaller compared to spring (Fig. 5 C-E), average growth was higher in autumn and winter than in spring (Fig. 12).

## 4 Discussion

### 4.1 Water chemistry

The aim of the study was to detect influences of elevated $pCO_2$ and temperature on growth and destruction of calcareous tubeworm shells under near-natural conditions in different seasons. The temperature manipulations produced consistent offsets of 4-5°C between the respective treatments (Fig. 3). However, the basin water acidification (pH, saturation state $\Omega$) induced by elevated $pCO_2$ was more complex. The average pH and $\Omega$ values were highest in the control treatments and lowest in the $+CO_2+T$ treatments. Intermediate values occurred in the $+CO_2$ and $+T$ treatments. At the same $pCO_2$ level pH was lower in the elevated temperature treatments (Table 1; Wahl et al., 2015). This was probably caused by biological activity or nutrient cycling. It cannot be explained by the carbonate chemistry, which would result in higher pH at elevated temperatures under otherwise constant conditions (Lewis and Wallace, 1998). The mean pH difference between the $+CO_2+T$ and the control treatments was 0.2 units in summer and autumn and 0.4 units in spring and winter (Tab. 1). This simulated pH change is in good agreement with the predicted pH decrease at the end of this century (Omstedt et al., 2012; IPCC, 2014).

The seasonal fluctuations of pH (0.4 to 0.6) and $\Omega$ (0.9 to 1.9) exceeded the respective differences between treatments (pH: 0.2 to 0.4, $\Omega$: 0.2 to 1.2). This has to be considered when comparing data from different seasons (e.g. Fig. 13). In addition, the strong diurnal cycles of pH ($\leq 0.6$) and saturation ($\leq 2.2$) complicate interpretations of carbonate chemistry impacts on tube growth and corrosion (e.g. Fig. 10).

Such interpretations are further hampered by potential impacts from the diffusive boundary layer (DBL) forming at the surface of *Fucus*, the substrate of *S. spirorbis* tubes (Spilling et al., 2010; Wahl et al., 2016). Photosynthetic activity during the day can elevate pH and saturation state in the algal DBL compared to the bulk fluid. Average saturation of the autumn experiment was as low as $\Omega=0.6$ in the bulk fluids of some treatments. To elevate saturation from this value to slight oversaturation ($\Omega=1.1$) pH has to be increased by log(1.1/0.6), i.e by about 0.3 pH units. A pH elevation of this magnitude was reported by Wahl et al. (2016) at a DBL thickness corresponding to the height of *S. spirorbis* tubes. However, these observations were made in stagnant water while conditions in the benthocosms were quite turbulent due to artificial waves generated every 2 minutes (Wahl et al., 2015). Additionally, considering that light dose (light intensity and length of day, see Section 2.5) was reduced during the autumn experiment, it appears unlikely that photosynthesis-driven daytime pH elevation (Fig. 4) was sufficient to overcome under-saturated water conditions in the DBL. This means that *S. spirorbis* was able to build tubes with above-average rates (Gr



of ∼5 mm, Tab S2; Fig. 12) in spite of constant under-saturation ($t_{\Omega<1}$=100%) in the autumn treatments A1, B1 and E2 (Tab. 1). *S. spirorbis* tube growth in under-saturated water was previously observed by Saderne and Wahl (2013).

## 4.2 Reproduction and life cycle

*S. spirorbis* reproduces and releases larvae predominantly during the warm seasons (Knight-Jones and Knight-Jones, 1977;

Seed et al., 1981). Larvae settle in episodic pulses that may be coupled to fortnightly lunar or tidal cycles (De Silva, 1967; Daly, 1978). The episodic larval settlement provides an explanation for the presence of distinct populations in our experiments (Fig. 5 B-E). In line with previous studies, we found living (actively calcifying) juveniles at the beginning of all four seasonal experiments, i.e. in January, April, July, and October. This indicates that the Eckernförde Bay *S. spirorbis* population reproduces throughout the year, although, in January and April juveniles were very rare. At the end of the summer and autumn experiments

(September and December, respectively) we found numerous unstained living *S. spirorbis* specimens on the *Fucus* thalli which had shell diameters <1.3 mm. These juveniles obviously had settled during the experiments, indicating continuous reproduction in the benthocosms from July to December.

In addition to temperature, fecundity of *S. spirorbis* is affected by salinity and food supply and increases with individual size and age (Daly, 1978; Kupriyanova et al., 2001). Salinity fluctuated strongly during the experiments (from ∼10 psu in June to

∼20 psu in November-January, Fig. S2), but our data do not allow to draw conclusions about salinity impacts on reproduction. Food supply for the filter-feeding Baltic tubeworms is generally lower in winter and increases when increased light availability promotes phytoplankton growth in spring, summer and autumn. Juveniles were rare in the initial populations in April, when the water temperature was still <10°C and in January when temperatures had decreased to <5°C (Fig. 3). In April, however, phytoplankton biomass is already high in the Kiel Bay area (Rheinheimer, 1996). Therefore, temperature probably dominates

over food availability in controlling *S. spirorbis* reproduction in Eckernförde Bay.

## 4.3 Microstructures

The *S. spirorbis* investigated in this study display the typical chevron lamellae microstructure (Fig. 6) that has been reported from many serpulid species (e.g., Wrigley, 1950; Hedley, 1958; Burchette and Riding, 1977; Weedon, 1994; Buckman, 2015), including *Spirorbis spirorbis* (Ippolitov and Rzhavsky, 2015). We observed a complex 3-dimensional shape of the *S. spirorbis*

chevron lamellae with convex-forward curving layers that show convex-upward curving substructures (Fig. 8).

*S. spirorbis* tube walls are purely calcitic and two-layered with an irregularly oriented prismatic (IOP) ultrastructure in the chevrons of the wall's core and a spherulitic prismatic ultrastructure (SPHP) of the thin outer wall region (Ippolitov and Rzhavsky, 2015; Vinn et al., 2008). The IOP chevrons and the SPHP structure are also common in a range of other serpulid genera (*Crucigera*, *Floriprotis*, *Pyrgopolon*, *Spiraserpula*; Gee and Knight-Jones, 1962; Vinn, 2011). Weedon (1994) pointed

out that this complex internal tube architecture is difficult to explain with simple pasting models for serpulid calcification, i.e. secretion of calcium carbonate granules or of a mucus paste with small calcite crystals that are molded into the calcitic tube. It is likely that extracellular organic matrices and scaffolds play a role in tubeworm biocalcification (Tanur et al., 2010). Thin





layers of organic matrix could be secreted onto the surface of the growing shell, as indicated by the chevron-like pores between growth lamellae (Fig. 6b).

Chevron-like accretion of new tube lamellae is indicated by the shape of the shell's stain line (Fig. 7). The figure additionally shows that synchronously with the accretion of new chevron lamellae new material was added in a thin layer to the inner tube
wall. This wall thickening is in agreement with the observed tapering of the tube walls near the tube mouth (Fig. 6a, b). The asymmetric chevron lamellae structure of the *S. spirorbis* shells reported here (Fig. 7) has not been recorded previously in serpulid tubes. It shows that the inner and outer tube wall linings are differently constructed. *S. spirorbis* prefer to consolidate the inner surface of the tube while constructing new chevron layers.

In many *S. spirorbis* specimens the chevron lamellae of the central tube wall became visible as ring structures when the outer
tube wall layer broke off or dissolved (Fig. 9). The outer tube wall layer appears to be susceptible to corrosion in spite of its massive densely calcified nature (Fig. 6).

### 4.4 Shell corrosion

In a recent study Saderne and Wahl (2013) incubated *S. spirorbis* in a laboratory experiment for 30 days at three different pCO$_2$ levels (450 µatm, $\Omega$=1.8; 1200 µatm, $\Omega$=0.8; 3150 µatm, $\Omega$=0.4). They used specimens from the same site as in the
current study, i.e. from Eckernförde Bay. The tubes exhibited substantial dissolution at the highest pCO$_2$ condition (3150 µatm, $\Omega$=0.4), but not in experiments with lower pCO$_2$, even though waters were slightly undersaturated with respect to *S. spirorbis* calcite (1200 µatm, $\Omega$=0.8). In contrast, in the current study corrosion of *S. spirorbis* shell surfaces was common (>10% of the shells) when average seawater saturation state was below ∼0.9, indicating starting corrosion even at mildly undersaturated conditions. Shell corrosion increased with decreasing saturation when the seawater was undersaturated ($\Omega$<1; Fig. 10), but
occurred in only a few experiments for $\Omega$>1. It was completely absent at $\Omega$>2.

The *S. spirorbis* tubes in our experiments may have been more susceptible to shell corrosion compared to those of Saderne and Wahl (2013) for several reasons. First, the duration of the benthocosm experiments was much longer (>70 days) than the laboratory experiments (30 days). Second, during our experiments saturation state in the benthocosms fluctuated strongly between day and night (Fig. 3). Even with a mean saturation of 1 the shells may have been exposed to strongly undersaturated
water during night time. Saderne and Wahl (2013) did not record pH or $\Omega$ on diurnal timescales, but the low biomass (1 g per 0.6 L flask) and constant vigorous gas bubbling most likely prohibited strong diurnal fluctuations of saturation states in their experiments. Third, with the more natural conditions in the benthocosms (unfiltered seawater, presence of natural fauna and flora) bioerosion by microbes may have fostered corrosion of the shells. Therefore, our experiments indicate that under natural conditions *S. spirorbis* can be significantly affected by shell corrosion at acidification levels expected for the end of the century.
Corrosion in the mostly under-saturated waters of all autumn and the high-CO$_2$ winter experiments (t$_{\Omega<1}$>70%; Tab. 1, Fig. 10) was likely induced by mineral dissolution. In contrast, during the summer experiment when waters were mostly over-saturated (t$_{\Omega<1}$<50%) the tubes were affected by bioerosion. Boring organisms play an important role in the ecology of many marine habitats (Warme, 1977). Microborings were observed in a *S. spirorbis* shell from the summer control experiment. They probably affected the stability of the worm tube (Fig. 11). The few tubes recovered from residual *Fucus* thalli in the



summer experiments with elevated temperatures (+T and +CO$_2$+T) were mostly broken and strongly corroded (Fig. 16). These observations hint at a detrimental influence of elevated summer temperatures on *S. spirorbis* shells, either directly by affecting the worm's metabolism or indirectly through the reduction of grazing organisms (Werner et al., 2016). Additionally, irreversible damage of *Fucus* algae at high summer temperatures (>27°C, Graiff et al., 2015a) leads to substratum loss for *S. spirorbis*,

which preferentially settle on *Fucus* (De Silva, 1962; O'Connor and Lamont, 1978).

    *S. spirorbis* tubes that grow during the warm season might be especially susceptible to mechanical stress and bioerosion. As shown in Section 3.5, summer tubes were significantly lighter than expected for their size. This indicates thinner and/or less dense tube walls compared to autumn and spring specimens. With the increasing frequency and duration of summer heat waves in Central Europe predicted for the 21$^{st}$ century (Beniston et al., 2007; Gräwe et al., 2013), increased bioerosion and loss of

substratum can severely affect future *S. spirorbis* populations in the Baltic Sea.

## 4.5 Growth rate

Tube growth rates of *S. spirorbis* in our experiments were strongly controlled by the ontogenetic development. Growth rates were highest for juveniles and decreased when the worms got older and the tubes reached the maximum diameter range (Fig. 12). Similar growth-age relationships were found previously for *S. spirorbis* and other serpulid worms (O'Connor and Lamont,

1978; Kupriyanova et al., 2001; Riedi, 2012).

    However, if only adult specimens of similar initial sizes are considered, *S. spirorbis* tubes grew more rapidly in autumn and winter (October - March) than in spring (April - June), with the slowest growth occurring in the adult summer population (July - September, Fig. 12). We know of no comparable published data. Reports of more rapid tube growth in summer compared to winter for several temperate serpulid species (Riedi, 2012; Kupriyanova et al., 2001) usually refer to the ontogenetic effect

described above, i.e. enhanced growth of juvenile serpulids. The enhanced cold season growth of adult *S. spirorbis* in our experiments was quite unexpected. Water was frequently under-saturated with respect to Mg-calcite during autumn and winter (48 to 98 % of experimental time, compared to 9 to 43 % in spring and summer, Table 1, Fig. 3). Other factors may play a role, like food availability and salinity. Food supply is generally lower in winter than during the warm seasons when increased light availability promotes phytoplankton growth. It consequently provides no explanation for the observed enhanced cold

season growth of adult *S. spirorbis*. Salinity was high during November-January and lowest in June. Spring salinities between 10 and 15 psu contrasted with autumn and winter values between 16 and 21 psu (Fig. S2). Enhanced calcification at higher salinities was previously observed in Baltic bivalves (Hiebenthal et al., 2012) and may potentially provide an explanation for enhanced growth of adult *S. spirorbis* during the autumn and winter experiments. No significant treatment effects on growth were detected (see below). Consequently, the cold season growth enhancement of adult *S. spirorbis* is not an artefact

of increased temperatures and pCO$_2$ levels in the benthocosms. We suggest that, in addition to possible salinity effects, the reduced growth of adult specimens during the warm seasons reflects enhanced reproductive activity during this time (Knight-Jones and Knight-Jones, 1977; Seed et al., 1981), re-allocating energy resources from calcification to reproduction and thus reducing tube growth capacities.





The influence of increased $pCO_2$ on the growth of *S. spirorbis* worm tubes was previously studied in the experiments of Saderne and Wahl (2013). A significant growth rate reduction was only observed for adult specimens at the highest $pCO_2$ (3150 µatm, $\Omega$=0.4). No significant growth rate reduction was found at the intermediate $pCO_2$ level of 1200 µatm. In agreement with these results we found no significant change in tube growth parameters ($Gr$, $Gr/D_i$, $D_f/D_i$) when elevating $pCO_2$ from ambient

levels to 1100 µatm (Fig. 14), corresponding to average saturation values as low as $\Omega$=0.6 (Tab. 1). We detected no significant impact of saturation state on growth (tube length or diameter) in any season. However, the average weights of newly grown tubes correlated with saturation states in the spring and autumn experiments (Fig. 13).

Apparently, the Baltic *S. spirorbis* worms are able to build their tubes with little changes in extension rates at $pCO_2$ levels as high as 1100-1200 µatm. Notably, these $pCO_2$ values are in the range of their natural habitats (385 to 2500 µatm; Thomsen

et al., 2010, 2013; Wahl et al., 2015). However, the tubes that are formed at lower $CaCO_3$ saturation may be more fragile. This is in line with results from cultured juvenile worm tubes of the tropical serpulid species *Hydroides elegans*, which showed reductions in shell hardness and wall thickness at lowered pH and $CaCO_3$ saturation states (Chan et al., 2012, 2013; Li et al., 2014).

As discussed in Section 4.4 high temperatures in the +T and +$CO_2$+T treatments of the summer experiment (average T of

24°C, Tab. 1) led to high mortality and strongly reduced growth of *S. spirorbis* tubes (Fig. 16). The only other significant temperature influence on growth was found in the juvenile populations of the autumn experiment. The higher temperature in the +T treatment induced higher growth rates of the juvenile populations compared to the control treatment (Fig. 14). There was, however, no significant temperature influence on the growth parameters at elevated $pCO_2$ (+$CO_2$+T treatment), possibly indicating interactions between the effects of temperature and $pCO_2$ on growth.

## 5   Conclusions

The results of our benthic mesocosm experiments clearly demonstrate that the growth of *S. spirorbis* tubes is predominantly controlled by ontogenesis. Elevated $pCO_2$ levels, lowered pH and calcium carbonate saturation states expected for the end of the 21[st] century had no significant impact on tube extension rates. Rather, *S. spirorbis* is capable of calcifying in water under-saturated with respect to the Mg-calcite of its shell. New tube parts were observed to be formed in under-saturated water

when at the same time parts of the older tube were being corroded (Fig. 17). This is clear evidence for a strict biological biomineralization control of *S. spirorbis*.

Opposed to the batch culture experiments of Saderne and Wahl (2013) significant shell corrosion occurred in our experiments at a $pCO_2$ of 1100 µatm. While acidification had no impact on shell extension, shell corrosion increased with progressing acidification and under-saturation. Additionally, increased bioerosion, reduced growth, and loss of substratum occurred at high

summer temperatures. Most *S. spirorbis* were not able to survive at a mean temperature of 24°C in the benthocosms. On the other hand, among the juvenile populations of the autumn experiment, elevated temperatures (15°C) increased tube growth rates, but only under ambient $pCO_2$ conditions.



We conclude that under continued warming and ocean acidification, with conditions expected for the end of the 21$^{st}$ century, *S. spirorbis* in the Baltic Sea could be seriously affected by high summer temperatures and by enhanced dissolution and bioerosion in increasingly warmer, acidified seawater. These results contrast with previous batch culture experiments, indicating the need for experiments simulating near-natural conditions in climate change research.

5   *Competing interests.*  The authors declare that they have no conflict of interest.

*Acknowledgements.*  Many thanks go to Björn Buchholz, Fin Ole-Petersen, Martin Wahl and all members of the BIOACID II consortium "Benthic Assemblages" for setting up and maintaining the Kiel Outdoor Benthocosms. Vincent Saderne, Esther Rickert and Nele Wendländer provided invaluable help for collecting, handling and studying *Spirorbis*. Stefan Krause, Mario Thöner and Manuela Goos helped with preparation and imaging of polished samples. Mark Lenz and Stephanie Schurigt kindly helped with the intricacies of statistical analyses.

10   Iris Schmiedinger was of invaluable help in the laboratory during water chemistry analyses. This study was funded by the collaborative project BIOACID Phase II of the German Federal Ministry of Education and Research (BMBF; FKZ 03F0655F) and by Leibniz IOW.




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




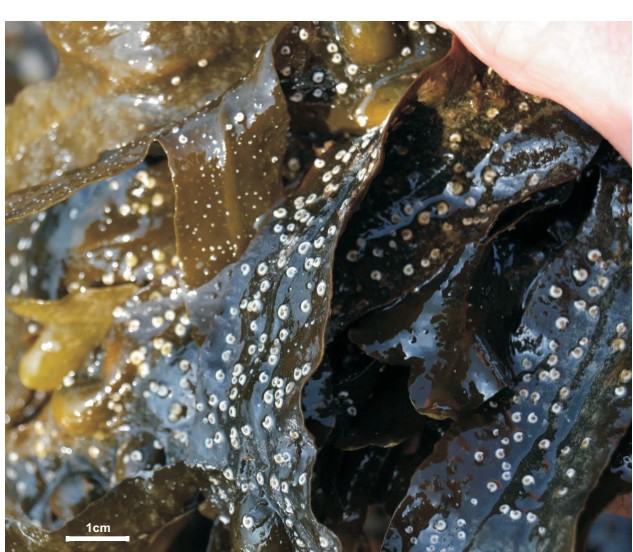

**Figure 1.** *Spirorbis spirorbis* specimens attached to living brown alga *Fucus vesiculosus*. Juvenile (white dots) and adult (white spires) specimens of *S. spirorbis* are visible.




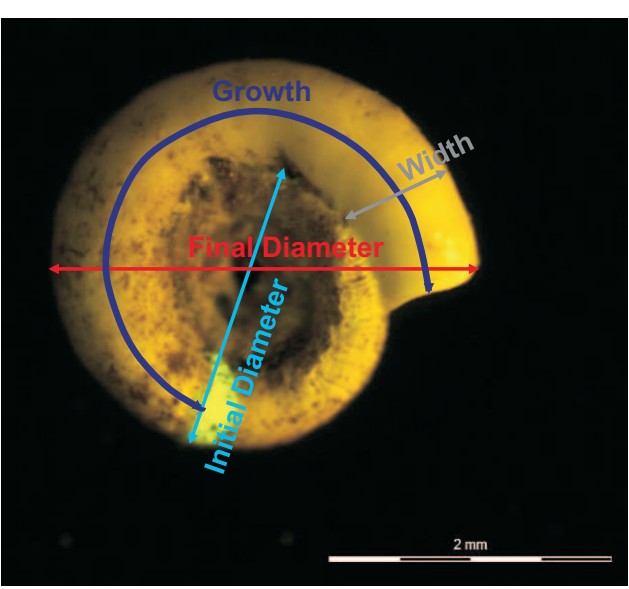

**Figure 2.** Fluorescence microscope view of *S. spirorbis* shell with indicators for measured size parameters used in this study (initial diameter, final diameter, growth length, tube width). Brightly fluorescent yellowish calcein stain line (lower left) marks beginning of shell part grown during experiment.





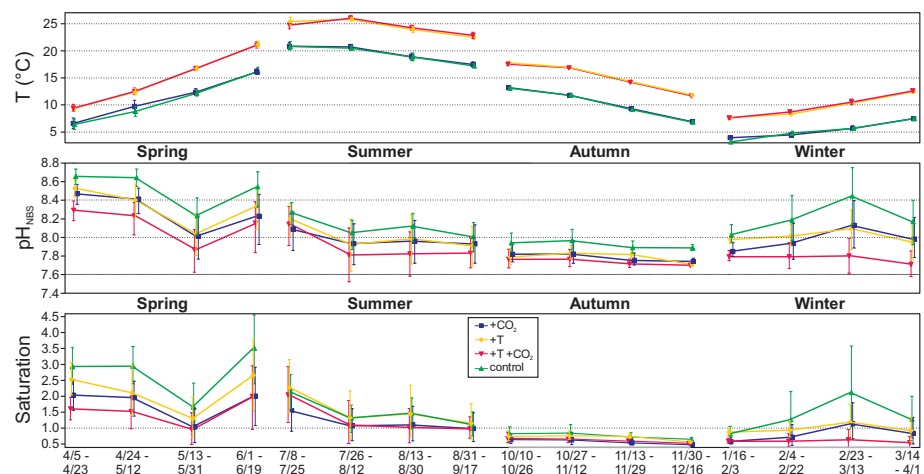

**Figure 3.** Average water temperature, pH and saturation state with respect to aragonite (as proxy for *S. spirorbis* Mg-calcite) in the four different treatments. Each of the four seasonal experiments is divided into four sub-periods lasting 17-19 days (start and end dates indicated at x-axis). Error bars indicate minimum and maximum values of the mean diurnal cycle during the sub-periods.





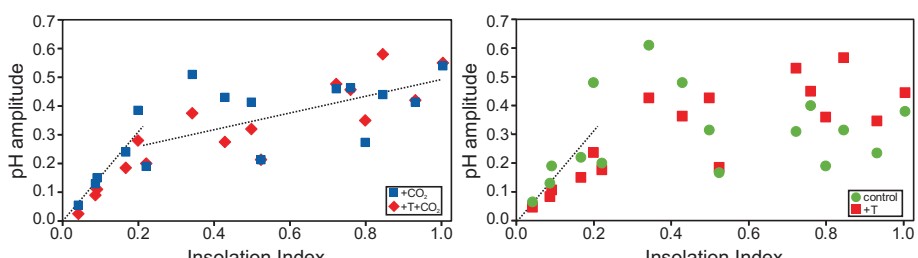

**Figure 4.** Light dependence of diurnal pH cycles. Average diurnal pH amplitudes in the benthocosm basins for $CO_2$-enriched (left) and ambient treatments (right) plotted versus the simple insolation index for the sub-periods of the four seasonal experiments. All treatments show a strong light dependence at low light levels (<0.2) but only a weak trend for the $CO_2$-enriched treatments at stronger insolation. Dotted trend lines indicate significant linear regressions ($p<0.05$).




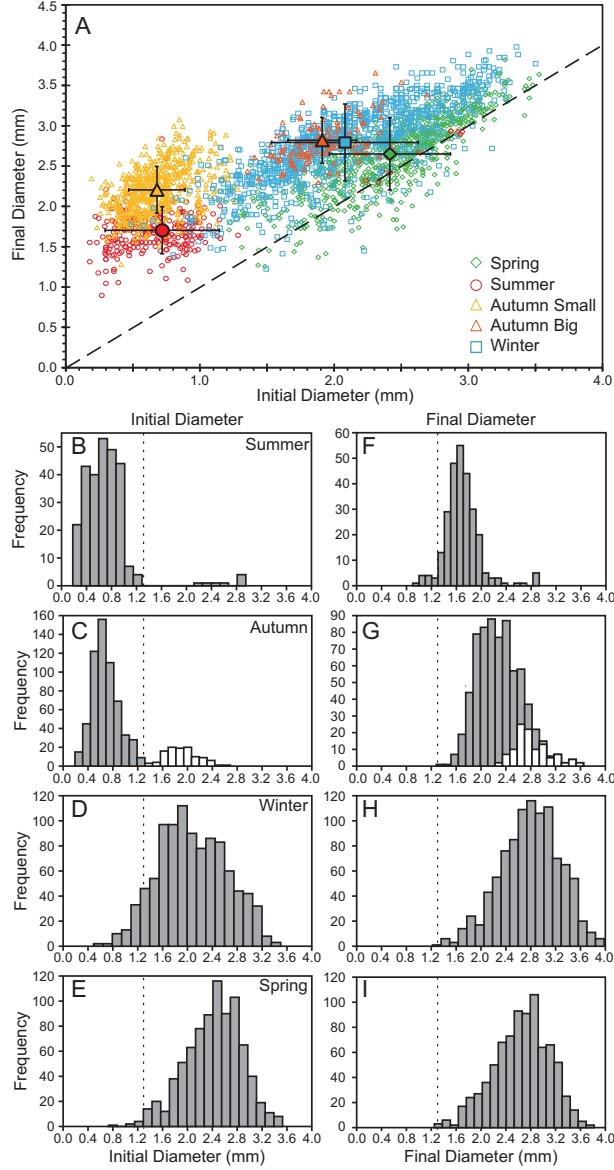

**Figure 5. A:** Cross-plot of initial and final diameters of individual *S. spirorbis* tubes from all treatments and seasons. Note that only specimens that grew during the initial staining were included. Big symbols are seasonal mean diameters (±standard deviation). The population of the autumn experiment was subdivided into a small and a big sub-population as described in the text. The tubes that plot above the dashed diagonal line of non-growth did grow during the experiments. Note that most specimens of the spring and winter experiments plot close to this line indicating little growth. Specimens below the line showed deformations and irregular growth. **B-E**, **F-I:** Size distributions of *S. spirorbis* at the start and end of the four seasonal experiments, respectively. Vertical dashed lines at 1.3 mm indicate threshold diameter between "juvenile" and "adult" populations. White bars in C and G represent the autumn-big population. Frequency indicates number of specimens in each size class.

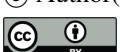



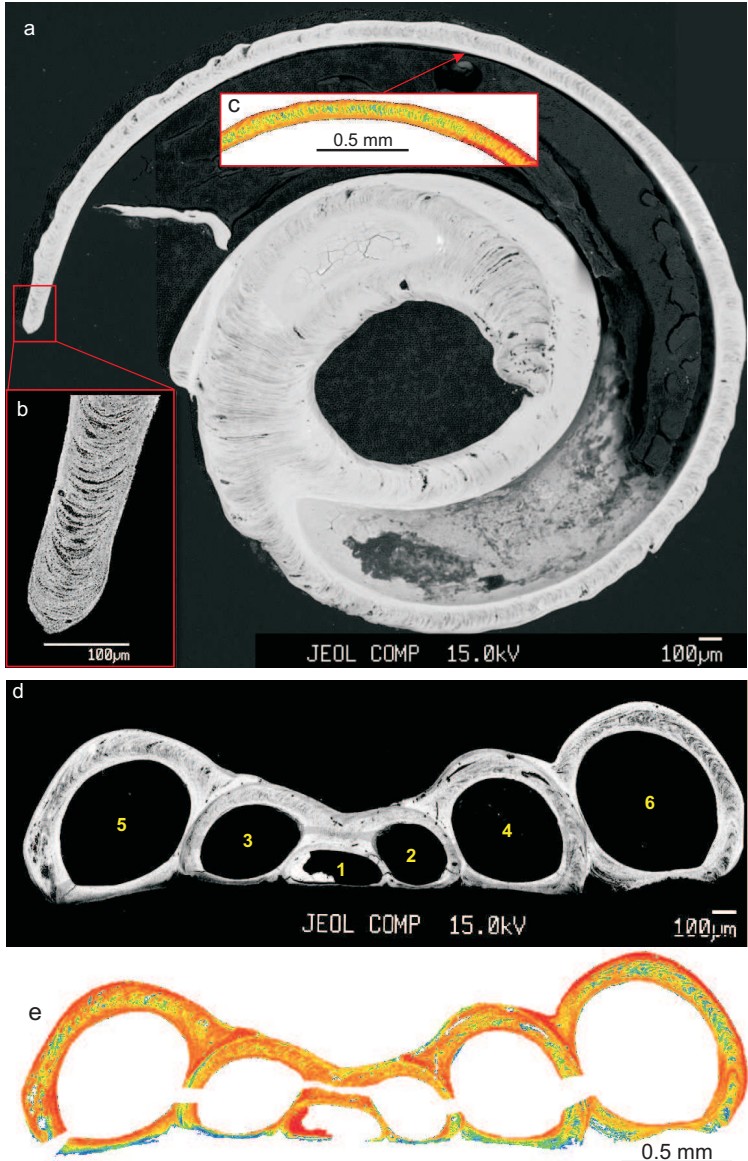

**Figure 6.** Backscatter SEM images (BEI) and electron microprobe (EMP) calcium maps of embedded and polished *S. spirorbis* specimens from the Winter control experiment. **(a)** cross section viewed from shell bottom. **(b)** detail from area in red frame, outer wall at the tube mouth showing convex forward lamellae (chevron structure). **(c)** EMP calcium map of upper right tube wall showing densely calcified outer layers along the inner and outer rim (red, high calcium concentration). Inner parts of tube wall are laminated and less calcified (yellow-green, low calcium concentration). **(d)** longitudinal cross section. Numbers indicate order of tube whorls. 1: juvenile tube, partly filled with secondary dense material (left). 6: latest whorl, added during the experiment. **(e)** calcium EMP map showing densely calcified wall rims and laminated less calcified wall fillings. Later whorls partly coat the older tube with a thick calcium-rich shell layer. White areas cutting the tube walls are artefacts of data acquisition.





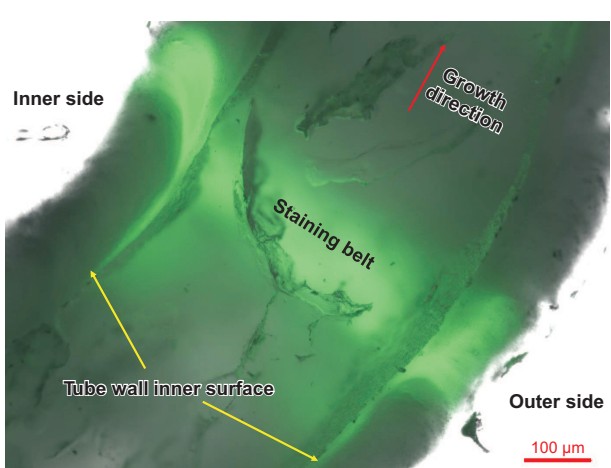

**Figure 7.** Fluorescent microscope image of *S. spirorbis* tube cross section with staining line showing green fluorescence. Sample from Spring +CO$_2$+T experiment. The green belt is the stained part of the shell that formed during the 3 days of calcein staining before the start of the experiment. Newly grown shell forms a lining along the inner tube wall surface (yellow arrows). Red arrow indicates the growth direction.





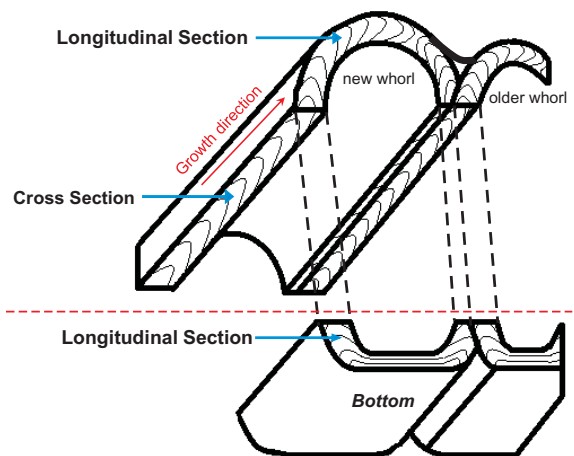

**Figure 8.** Schematic line drawing of the shell structures of *S. spirorbis* showing the orientations of cross and longitudinal sections (blue arrows) and the respective orientation of the chevron lamellae. Red arrows indicate growth direction. The parts below the red dashed line are only visible in the longitudinal sections. Tube wall thickness is about 100 μm.





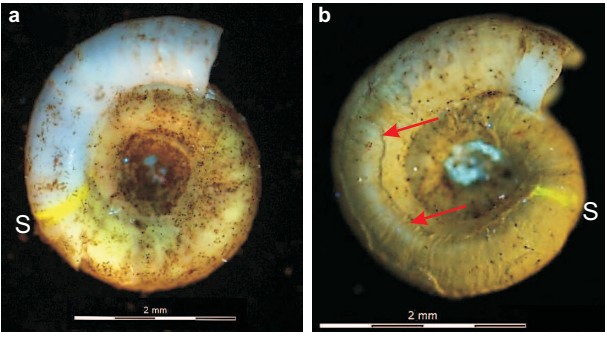

**Figure 9.** Pristine and corroded *S. spirorbis* shells. **(a)** Pristine smooth surface without visible corrosion. Specimen from winter control experiment. **(b)** Corroded surface of a tube from winter +CO$_2$+T experiment. The outermost shell layer was removed by corrosion, exposing the ring structure of the underlying shell layer (arrows). "S" indicates position of the stain line (start of experiment).





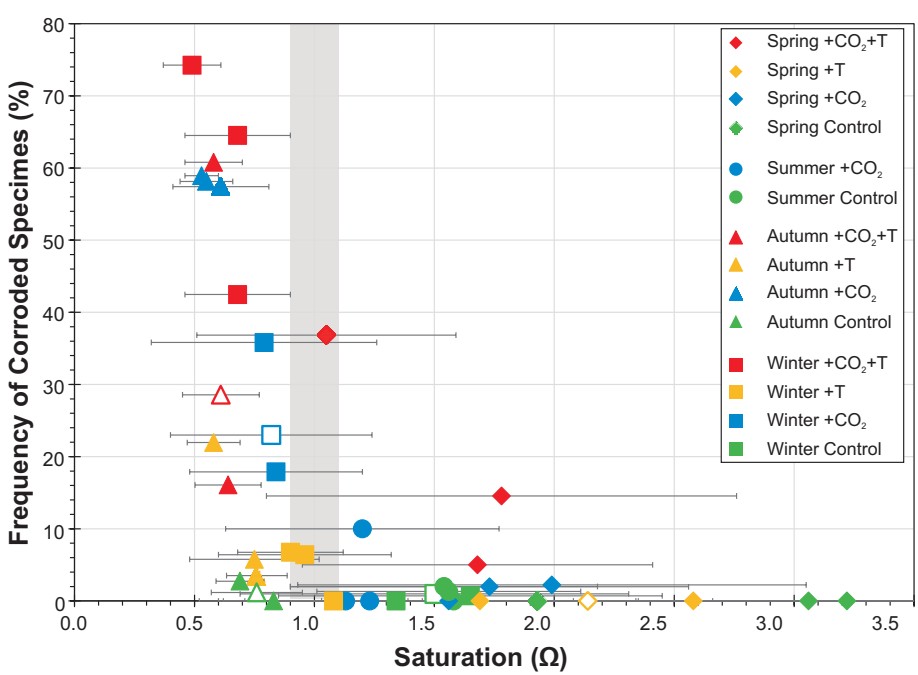

**Figure 10.** Proportion of corroded samples as a function of the calcium carbonate saturation state of seawater. Each data point represents one basin. Grey bar indicates saturated water ($\Omega=1.0\pm0.1$). Error bars are standard deviations of saturation data for each basin (Table 1). For basins without available saturation data the treatment averages were used (open symbols).





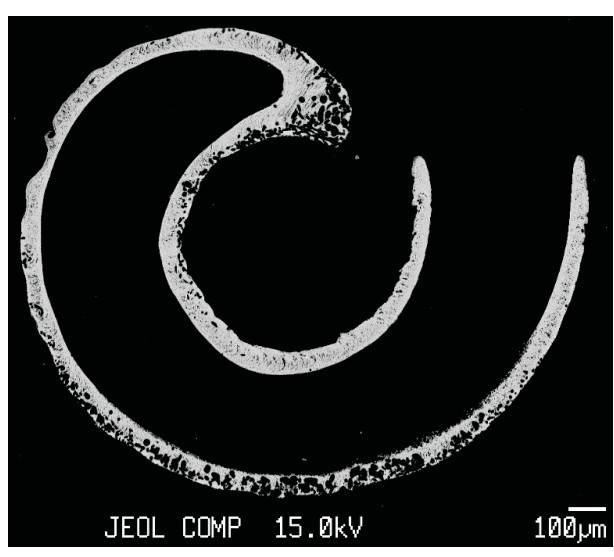

**Figure 11.** SEM image (BEI) of polished cross section of *S. spirorbis* shell from summer control experiment. Dark spots are microborings mostly affecting the outer tube wall.





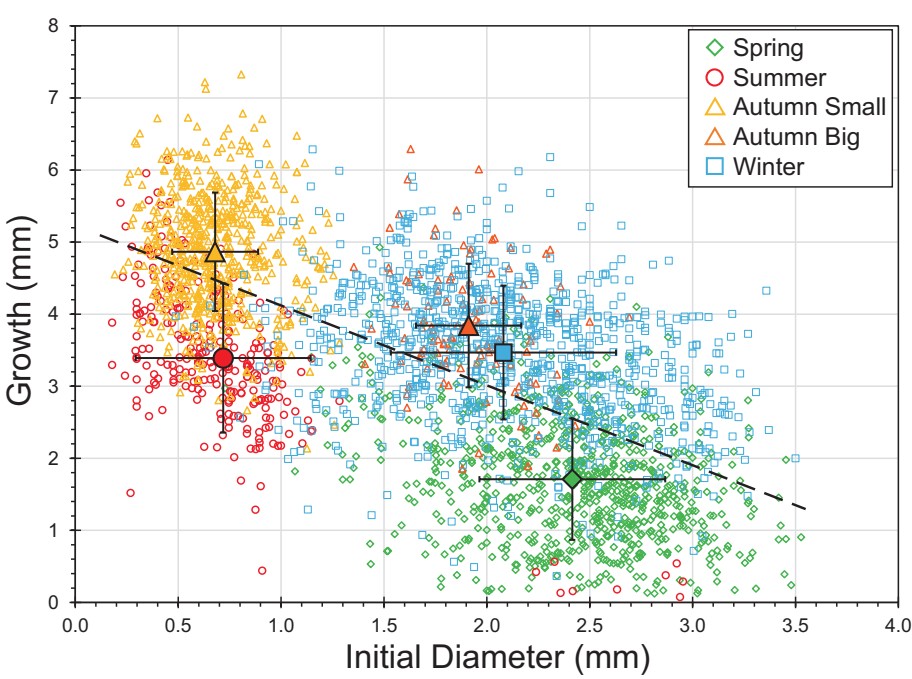

**Figure 12.** Length of new tube growth during the experiments plotted against initial diameter. Dashed line is a linear fit to the data ($R^2$=0.41, n=2783, p=0). Data are from all experiments and treatments. Small symbols indicate individual *S. spirorbis* specimens, bold symbols are seasonal mean values (±1 standard deviation). Autumn small and big populations are plotted separately.





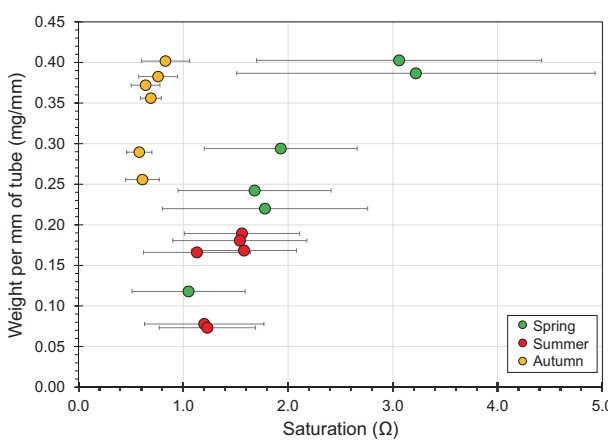

**Figure 13.** Weights of *S. spirorbis* tube segments that grew during the spring, summer and autumn experiments. Data points are average weights per millimeter of tube of selected basins (Table S1), plotted against average saturation state. Vertical error bars (±1 % of measured weight) are smaller than the symbols. Horizontal error bars (±1 standard deviation) represent the variability of saturation in each basin during the experiments.




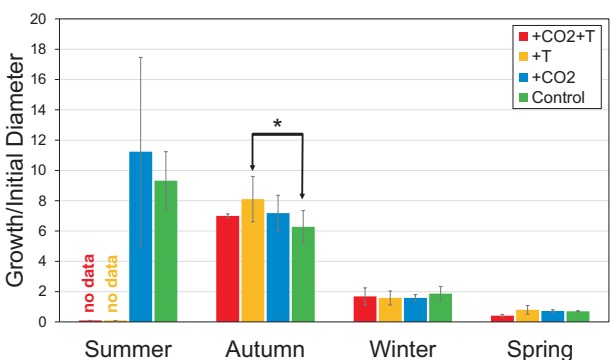

**Figure 14.** Average growth (Gr/$D_i$) in different treatments during seasonal experiments. In autumn, growth differed significantly between the +T and control treatments. The effect is only seen in the juvenile specimens, which however dominate the autumn population. In summer, no tubes were recovered from the elevated temperature treatments. Results from three-way ANOVA and Tukey's HSD tests; * significant difference ($p < 0.05$).





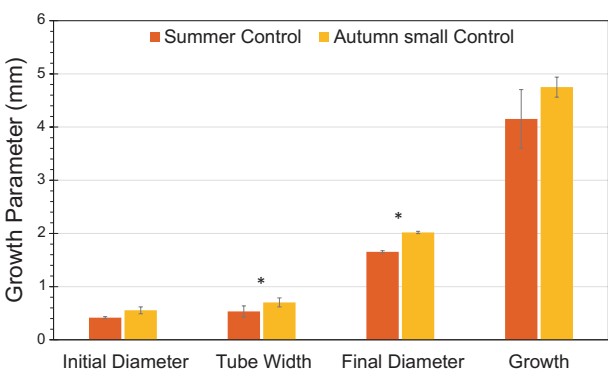

**Figure 15.** Size and growth of juvenile populations in control treatments of summer and autumn experiments. Two-way ANOVA and Tukey's HSD tests; * significant difference (p <0.05).





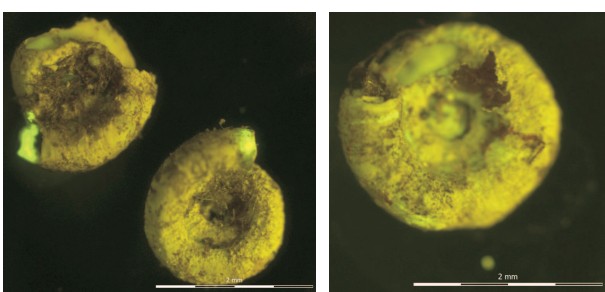

**Figure 16.** Strongly bio-eroded and broken *S. spirorbis* tubes from summer +T experiment (left: basin A2, right: basin C2). Tubes are partly covered by filamentous algae. Note spongy appearance of tubes due to intense microboring. Stain lines indicating the start of the experiment are visible at the tube mouths in the left picture. Scale bars are 2 mm.





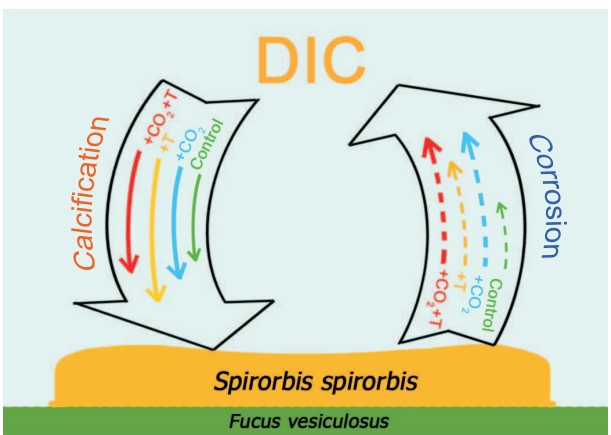

**Figure 17.** Synchronous calcification and shell corrosion in the under-saturated waters (mean $\Omega<0.8$) of the autumn experiment. Length of arrows indicates relative magnitudes of treatment influences on shell calcification (left, solid arrows) and shell corrosion (right, dashed arrows), respectively. Shell growth showed little variability between treatments, except for increased growth in the +T treatment. Corrosion was strongly increased in the high-$CO_2$ treatments and slightly enhanced in the +T treatment. DIC: dissolved inorganic carbon.




**Table 1.** Average water data of the four treatments in the four seasonal experiments.

| | | Spring (April - June 2013) | | | | Summer (July - September 2013) | | | |
|---|---|---|---|---|---|---|---|---|---|
| | | A/B | C/D | E/F | Mean | A/B | C/D | E/F | Mean |
| +T+CO$_2$ | T (°C) | 14.7±4.6 | 14.5±4.6 | 14.4±4.7 | 14.5±0.2 | 24.3±1.9 | 24.2±2.0 | 24.6±2.0 | 24.4±0.2 |
| | pH$_{NBS}$ | 7.98±0.27 | 8.23±0.30 | 8.22±0.28 | 8.1±0.1 | 7.87±0.24 | 7.91±0.23 | 7.94±0.27 | 7.9±0.04 |
| | Ω | 1.05±0.54 | 1.78±0.98 | 1.68±0.73 | 1.5±0.4 | 1.19±0.63 | 1.28±0.63 | 1.43±0.92 | 1.3±0.1 |
| | t$_{Ω<1}$ (%) | 51 | 24 | 20 | 32±17 | 47 | 42 | 39 | 42±4 |
| +T | T (°C) | 14.6±4.7 | | 14.5±4.7 | 14.6±0.1 | 24.3±1.9 | 24.2±2.0 | 24.3±2.0 | 24.3±0.1 |
| | pH$_{NBS}$ | 8.22±0.24 | no data[a] | 8.43±0.31 | 8.3±0.2 | 7.99±0.19 | 8.00±0.21 | 8.04±0.27 | 8.0±0.03 |
| | Ω | 1.69±0.66 | | 2.58±1.19 | 2.1±0.6 | 1.46±0.62 | 1.49±0.64 | 1.73±0.98 | 1.6±0.1 |
| | t$_{Ω<1}$ (%) | 16 | | 11 | 13±4 | 31 | 29 | 30 | 30±1 |
| +CO$_2$ | T (°C) | 11.1±3.7 | 11.1±3.8 | 10.6±3.9 | 10.9±0.3 | 19.3±1.8 | 19.4±1.8 | 19.3±1.8 | 19.3±0.1 |
| | pH$_{NBS}$ | 8.24±0.26 | 8.28±0.29 | 8.34±0.35 | 8.3±0.1 | 7.96±0.20 | 8.01±0.17 | 7.97±0.22 | 8.0±0.02 |
| | Ω | 1.56±0.78 | 1.73±0.83 | 1.99±1.06 | 1.8±0.2 | 1.13±0.51 | 1.23±0.46 | 1.20±0.57 | 1.2±0.1 |
| | t$_{Ω<1}$ (%) | 27 | 24 | 24 | 25±2 | 46 | 36 | 46 | 43±6 |
| control | T (°C) | 10.7±3.8 | 10.0±4.3 | 10.9±3.7 | 10.5±0.5 | 19.2±1.8 | 19.5±1.8 | | 19.4±0.2 |
| | pH$_{NBS}$ | 8.36±0.22 | 8.61±0.27 | 8.59±0.35 | 8.5±0.1 | 8.11±0.18 | 8.14±0.14 | no data[b] | 8.1±0.02 |
| | Ω | 1.93±0.73 | 3.06±1.36 | 3.22±1.71 | 2.7±0.7 | 1.54±0.64 | 1.58±0.50 | | 1.6±0.03 |
| | t$_{Ω<1}$ (%) | 11 | 6 | 11 | 9±3 | 24 | 13 | | 19±8 |

| | | Autumn (October - December 2013) | | | | Winter (January - March 2014) | | | |
|---|---|---|---|---|---|---|---|---|---|
| | | A/B | C/D | E/F | Mean | A/B | C/D | E/F | Mean |
| +T+CO$_2$ | T (°C) | 15.1±2.5 | 15.0±2.6 | | 15.1±0.1 | 9.8±1.9 | 9.8±2.1 | *11.7±1.1*[d] | 9.8±0.0 |
| | pH$_{NBS}$ | 7.71±0.08 | 7.76±0.09 | no data[c] | 7.7±0.04 | 7.70±0.11 | 7.84±0.15 | *7.84±0.14*[d] | 7.8±0.1 |
| | Ω | 0.58±0.12 | 0.64±0.14 | | 0.6±0.04 | 0.49±0.12 | 0.68±0.22 | *0.68±0.22*[d] | 0.6±0.1 |
| | t$_{Ω<1}$ (%) | 100 | 97 | | 98±2 | 100 | 89 | *87*[d] | 95±8 |
| +T | T (°C) | 15.1±2.5 | 15.3±2.5 | 15.2±2.6 | 15.2±0.1 | 9.6±2.0 | 9.8±2.1 | 9.4±2.0 | 9.6±0.2 |
| | pH$_{NBS}$ | 7.83±0.09 | 7.80±0.20 | 7.72±0.10 | 7.8±0.06 | 8.05±0.15 | 7.99±0.17 | 7.98±0.11 | 8.0±0.04 |
| | Ω | 0.76±0.13 | 0.75±0.27 | 0.58±0.11 | 0.7±0.1 | 1.08±0.37 | 0.96±0.36 | 0.90±0.22 | 1.0±0.1 |
| | t$_{Ω<1}$ (%) | 95 | 86 | 100 | 94±7 | 53 | 68 | 76 | 66±12 |
| +CO$_2$ | T (°C) | 10.3±2.7 | 10.6±2.5 | *7.8±1.2*[d] | 10.5±0.2 | | 5.2±1.7 | 5.4±1.5 | 5.3±0.1 |
| | pH$_{NBS}$ | 7.76±0.05 | 7.81±0.12 | *7.79±0.07*[d] | 7.8±0.04 | no data[b] | 7.95±0.22 | 7.99±0.18 | 8.0±0.03 |
| | Ω | 0.53±0.07 | 0.61±0.20 | *0.55±0.11*[d] | 0.6±0.06 | | 0.79±0.47 | 0.84±0.36 | 0.8±0.04 |
| | t$_{Ω<1}$ (%) | 100 | 93 | *99*[d] | 97±5 | | 78 | 74 | 76±3 |
| control | T (°C) | 10.2±2.6 | 10.4±2.6 | | 10.3±0.1 | | 5.2±1.6 | *6.9±1.1*[d] | 5.2 |
| | pH$_{NBS}$ | 7.88±0.06 | 7.95±0.11 | no data[c] | 7.9±0.05 | no data[b] | 8.20±0.24 | *8.30±0.24*[d] | 8.2 |
| | Ω | 0.69±0.10 | 0.83±0.23 | | 0.8±0.1 | | 1.34±0.82 | *1.65±0.80*[d] | 1.3 |
| | t$_{Ω<1}$ (%) | 99 | 81 | | 90±13 | | 48 | *28*[d] | 48 |

Mean values for temperature, pH, saturation (Ω) and percent of experimental time when basins were undersaturated with respect to aragonite and Mg-calcite (t$_{Ω<1}$).

Columns show mean values for single basins (A1, A2, B1, B2, etc.) and averages for each treatment with ±1sd ranges. a: pH data only for last 2 weeks of experiment; b:

no pH data recorded; c: no data recorded; d: data only for last 4 weeks of experiment. Data excluded from mean. **42**





**Table 2.** Corroded sample percentages (%).

| Basin | Spring | Summer | Autumn | Winter |
|---|---|---|---|---|
| A1 (+T+$CO_2$) | 36.8 | -- | 60.8 | 74.3 |
| A2 (+T) | 0.0 | -- | 3.5 | 0.0 |
| B1 (+$CO_2$) | 0.0 | 0.0 | 58.9 | 23.0 |
| B2 (control) | 0.0 | 2.0 | 2.7 | 1.0 |
| C1 (+T+$CO_2$) | 14.5 | -- | 16.1 | 64.5 |
| C2 (+T) | 0.0 | -- | 5.8 | 6.4 |
| D1 (+$CO_2$) | 2.0 | 0.0 | 57.4 | 35.8 |
| D2 (control) | 0.0 | 0.0 | 0.0 | 0.0 |
| E1 (+T+$CO_2$) | 5.0 | -- | 28.6 | 42.5 |
| E2 (+T) | 0.0 | -- | 22.0 | 6.7 |
| F1 (+$CO_2$) | 2.2 | 10.0 | 58.1 | 17.9 |
| F2 (control) | 0.0 | 1.3 | 1.2 | 0.7 |

No specimens were recovered from elevated temperature treatments in
summer



**Table 3.** Mean size and growth parameters of juvenile *S. spirorbis* populations in summer and autumn.

| Season | Basin | Count | $D_i$ (mm) | $D_f$ (mm) | Gr (mm) | TbWd (mm) |
|---|---|---|---|---|---|---|
| Summer control | B2 | 18 | 0.44±0.07 | 1.70±0.23 | 4.57±0.82 | 0.55±0.07 |
| | D2 | 19 | 0.40±0.08 | 1.53±0.20 | 3.53±0.64 | 0.51±0.08 |
| | F2 | 43 | 0.42±0.20 | 1.73±0.20 | 4.36±1.03 | 0.55±0.06 |
| Summer +CO$_2$ | B1[a] | 6 | 0.37±1.06 | 1.54±0.58 | 2.45±1.66 | 0.51±0.09 |
| | D1[a] | 3 | 0.18±0.07 | 1.46±0.12 | 3.88±0.44 | 0.45±0.08 |
| | F1[a] | 10 | 0.33±0.21 | 1.27±0.67 | 2.95±0.56 | 0.44±0.09 |
| Autumn control | B2 | 13 | 0.60±0.07 | 2.09±0.19 | 4.97±0.51 | 0.73±0.08 |
| | D2 | 19 | 0.48±0.09 | 1.93±0.21 | 4.62±0.67 | 0.69±0.09 |
| | F2 | 20 | 0.58±0.07 | 2.05±0.22 | 4.67±0.80 | 0.69±0.08 |
| Autumn +CO$_2$ | B1 | 28 | 0.54±0.09 | 2.06±0.18 | 4.41±0.49 | 0.76±0.07 |
| | D1 | 28 | 0.54±0.11 | 1.98±0.22 | 4.53±0.43 | 0.73±0.08 |
| | F1 | 30 | 0.54±0.11 | 2.19±0.21 | 5.20±0.55 | 0.76±0.07 |

Selected sub-populations with homogenous initial diameter range (similar median $D_i$) from summer and autumn control and +CO$_2$ treatments. Values are averages and standard deviations. $D_i$: initial diameter, $D_f$: final diameter, Gr: growth, TbWd: tube width. a: insufficient data to select sub-population with $D_i$ similar to other treatments.