# Peer review of "Effect of temperature rise and ocean acidification on growth of calcifying tubeworm shells (*Spirorbis spirorbis*): An *in-situ* benthocosm approach"

_Biogeosciences, 2017_

## Referee Comment (RC1) · VBSC Chan (Referee) · 11 Sep 2017

This manuscript entitled "Effect of ocean acidification and elevated temperature on growth of calcifying tubeworm shells (*Spirorbis spirorbis*): An in-situ benthocosm approach" authored by Sha Ni, Isabelle Taubner, Florian Böhm, Vera Winde, and Michael E. Böttcher examines the impacts of ocean acidification and warming on the calcifying tubeworm that grows on the Fucus algae in four seasonal experiments.

This study investigated the newly calcified materials of the tube worm using tube di-
mensions, net growth, microstructures under SEM, and electron microprobe detection of Ca density. The bethocosom setup provided naturally variating carbonate chemistry in seawater to the organisms for the experiment. This work suggests a seasonal trend of calcification related to maximum irradiance and the respective response to ocean acidification.

Unfortunately, these seasonal observations were not replicated and should be interpreted with caution.

Major comments:

This work has a lot of merits, for examples, the experiment was nicely controlled, the interaction of the study species with the algae is very interesting, the manipulated pH and temperature were both environmentally realistic, and the authors demonstrated a good level of understanding on the growth of tube. The tested levels of pCO2 and temperature treatments can help to better understand the impacts of Ocean Acidification on growth and calcification. However, the current statistical approach is prone to misleading conclusions on the effects of season and treatment. Firstly, the lack of sound independent replication season limits the conclusions that can be made. In addition, the approach in statistical analysis may overestimate the treatment effects from oversampling pseudoreplicates. Therefore, the authors should carefully modify their interpretations and make sure their observation is supported by sound statistics.

This work has only studied one year to understand potential seasonal impacts, however, authors tried to make a conclusion on seasonal impacts (i.e. Spring, Summer, Autumn, Winter) on the observed responses. This is not appropriate unless there were more than 3 years of observation to make each of spring, summer, autumn, and winter to have 3 independent and random observation. The authors are reminded that any seasonal effect observed is a suggestive trend for future experiments.

The interpretation of the result is not appropriate. The experiment is designed to appropriately answer whether there were effects by the 4 levels of treatment (ambient,

pCO2, +T, and pCO2+T), each benthocosm served as independent replicates for statistical analysis. HOWEVER, samples taken within the samples mesocosm are NOT independent replicates. I suggest the authors take the average value for every parameter they measure as the value in each replicate and perform 2-way ANOVA ore appropriate statistical test, take n=3. Over-sampling and pseudoreplication may lead to an overemphasis of the effects of treatment effects.

Discussion: This study evaluated the *S. spirobis* tubes in the natural growing scenarios in the presence of *Fucus* substrate. It is important to address a more general theory how calcifiers may benefit from the presence of photosynthetic activities of macroalga in the other ecosystems.

Furthermore, the impact of pCO2 on the photosynthetic activity of *Fucus* is also an important point of discussion. Here are a few studies which are relevant to this topic of discussion.

Bordeyne, F, Migne, A, Davoult, D. (2015) Metabolic activity of intertidal Fucus spp. communities: evidence for high aerial carbon fluxes displaying seasonal variability, MARINE BIOLOGY: 162 (10): 2119-2129 DOI: 10.1007/s00227-015-2741-6

Raddatz, S, Guy-Haim, T, Rilov, G, Wahl, M (2017) FUTURE WARMING AND ACIDIFICATION EFFECTS ON ANTI-FOULING AND ANTI-HERBIVORY TRAITS OF THE BROWN ALGA FUCUS VESICULOSUS (PHAEOPHYCEAE) JOURNAL OF PHYCOLOGY 53 (1): 44-58 DOI: 10.1111/jpy.12473

Detailed comments:

**Figure 5A, 12, 13, and 15**, should be removed or plotted again after inappropriate interpretation of seasonal response and pseudoreplication are eliminated.

**Page 6 Line 2** This selection of subpopulation needs to be further justified

**Page 12 Line 5-9** how did the researcher ensure the growth measurement of broken and strongly damaged tubes was accurate?

**Page 4 Line 9** "In this area,"

**Page 4 Line 23** How to quantify the volume of the blades, what has been done to ensure they are similar volume? How many worms were present in each replicate tank?

**Page 4 Line 26-27** provide the size/ volume of the boxes and duration of staining, at what density were the animal kept in the staining solution? Was the pH value before and after the staining monitored? Please provide this information for repeatability.

**Page 5 Line 11** "In total,"

**Page 6 Line 2** provide statistics for the difference

**Page 6 Line 16** list polishing materials and duration for reproducibility

**Page 6 Line 21** please provide excitation and emission wavelength, and resolution of images acquired

**Page 6 Line 27** pH calibration was at the lower range (pH 4 and pH 6.865), this requires extrapolation when measuring the ocean pH at 7.4-8.0. it is recommended to check the accuracy of this approach with another meter that has been calibrated by 3 points (for example pH 4, 7 10)

**Page 6 Line 31**, what is the importance of filtering and adding NaCl to TA samples? Please supplement the reason.

**Page 7 Line 1** should it be "Both DIC and TA measurements are calibrated. . ."?

**Page 7 Line 13** what was the technique used for measuring Ca Si and P?

**Page 7 Line 30** it is clearer to say " with respect to *S. spirobis* tube Mg-calcite"

**Page 9 Line 6-21** The use of adult and juvenile is not appropriate if the literature shows that sexual maturity begins at 1.9m. Authors are recommended to use neutral words such as small and large cohort/ young and old cohort

**BGD**

**Page 9 Line 22-29** Please explain the use of "mode" instead of "mean"

**Page 10 Line 15** Please reword and clarify the meaning of this line

**Page 10 Line 21** "Tube opening" not "tube mouth"

**Page 10 Line 24 and Figure 10** Please provide sample size that supports this percentage measurements

**Page 12 Line 23 to Page 13 Line 6** The seasonal effects is not supported by independent replication.

**Figure 1** please add arrows to show juvenile and adults

**Figure 2** Please make the font bigger

**Figure 7** Please improve the font choice for "Growth direction" and " Tube wall inner surface", it is not easy to read

**Figure 8** The labels for the longitudinal section and the cross section are mixed up, note that longitudinal section is the cut made along the long axis of the tube.

**Figure 9b** - please added annotation to indicate the microboring structures as mentioned in Page 11 Line 12

**Figure 12** use of red triangle and red circle makes the pattern hard to see, n=2783 is an example of pseudoreplication, please re-examine the result and make a new plot

**Figure 16** fonts for the scale bar are too small

**Table 2** - was is the total number of worms sampled for this percentage? Please provide the numbers by expanding the table.

---

## Referee Comment (RC2) · J. Bijma (Referee) · 27 Oct 2017

**General comments**

The paper by Ni et al. describes a very timely benthocosm approach to investigate the impact of temperature and CO2 on growth and calcification of a tubeworm. The location and the association of the tubeworms with seaweed leaves is intriguing as the Baltic Sea itself has a highly variable seasonal carbonate chemistry, on top of which the tubeworms experience a strong daily cycle of the carbonate chemistry in the diffusive

boundary layer on the leaves of the seaweed. The paper is within the scope of BG and the title clearly reflects the contents of the paper. It contributes new data, the authors give proper credit to related work and clearly indicate their own contribution. The abstract provides a concise and complete summary of the paper and the overall presentation of the manuscript is well structured and clear. In general, the language is fluent and precise. Because the paper is very comprehensive with lots of information and references, the authors could have considered writing two companion papers: One dealing with growth and population dynamics and another focussing on calcification and shell corrosion (please note this is just a remark and no requirement).

The description of the experiments and calculations are sufficiently complete and precise to allow their reproduction by fellow scientists. I suggest to add a picture of the benthocosm system. The scientific methods and assumptions are valid, clearly outlined and the results support the interpretations and conclusions. It should be mentioned, however, that mesocosm experiments, although closer to natural variability, also suffer from inherent drawbacks such as limited control compared to laboratory experiments. As such, the authors had no control on food availability, salinity nor day-length. This requires some additional discussion as outlined under "specific comments". Last but not least, the amount and quality of the supplementary material is appropriate.

Specific comments

If the selection of "Healthy F. vesiculosus plants bearing intermediate amounts of live S. spirorbis were collected for 4 seasonal experiments in less than 1.5 m water depth in Eckernförde Bay" was representative for the population in the field and all individuals on these leaves were considered for later analysis, fig. 5 is showing that reproduction occurs between spring and summer! This indicates that the authors are comparing two different cohorts that can have different sensitivities. Not just juvenile vs. adult but also generations affected by different starting (acclimation) conditions. This is known for physiological tolerance ranges (e.g. salinity) from organisms growing in different monsoon periods.

As stated before, mesocosm experiments suffer from limited control of certain parameters such as food availability, salinity or day-length. I wonder how well the applied normalisation e.g. for daylength ("simplified insolation index") works. I would appreciate some discussion about this as it is a critical seasonal parameter.

This maybe silly, but I wonder if the authors checked if all specimens were alive at the end of the experimental period?

I wonder if alkalinity changes were taken into account as the authors claim that "Enhanced calcification at higher salinities . . ... may potentially provide an explanation for enhanced growth of adult S. spirorbis during the autumn and winter experiments." Alkalinity scales with salinity. I suggest to provide full carbonate chemistry details.

Technical corrections

Fig. 4: I would expect to see a Michaelis-Menten type of kinetics. Please apply.

Fig. 9: Looks like the specimen in b) ("+CO2+T") grew ca. twice as much in length as the control under a). This seems counter-intuitive.

Fig. 14: Units on the y-axis are missing. In the caption it says "The effect is only seen in the juvenile specimens,. . ..". Please add that this is equivalent to the "autumn small population".
* * *

---

## Author Comment (AC1) · 7 Dec 2017

We thank the reviewer for his supportive comments and the constructive review of our manuscript. Below we respond to the reviewer's specific comments.

*Reviewer general comment 1*: Split manuscript. 1. growth and population dynamics. 2. calcification and shell corrosion.

**Response G1: We prefer to keep all of this information in a single, comprehensive manuscript because understanding ontogenetic growth and population dy-**

[Figure]

**namics is important for interpretations of calcification rates.**

*Reviewer general comment 2*: Add a picture of the benthocosm system.

*Response G2*: **A photograph showing the benthocosm setup will be added (see attached Fig. 1).**

*Reviewer general comment 3 and specific comment 2*: Mesocosm experiments have limited control and higher natural variability than laboratory experiments (e.g. food availability, salinity, day length). This requires additional discussion, specifically about the normalisation for day length (insolation index), which is a critical seasonal parameter.

*Response G3/S2*: **We agree with the reviewer that laboratory experiments are better controlled than in-situ mesocosms. However, the latter allow to consider the dynamics under near-natural boundary conditions needed to understand the reactions on manipulations like temperature and PCO2 under otherwise in-situ conditions. Only a comparison between laboratory and in-situ experiments can finally provide a solid frame to understand adaptations of natural communities to the expected changes in coastal ecosystems.**

**We will replace the "simplified insolation index" by the daily insolation sum (in kWh/m2) measured at the Meteorological Station of Geomar, which is situated very close to the benthocosm site (Fig. 2). This parameter implicitly includes day-length. Please note that we used this parameter only to explain the measured diurnal pH and saturation state changes and their variations between different seasons. We did not relate tube worm growth directly to insolation, but compared it to measured pH.**

**On the other hand, the strong seasonal variability makes it very difficult to compare the growth rate results between different seasons. This is discussed in Section 4.5 of our manuscript. Further possible conclusions about seasonal im-**

pacts on Spirorbis growth are limited by the one-year duration of our study. We have no seasonal replicates from different years to compare.

The following paragraph discussing the issues of limited environmental control will be added to the manuscript (Section 4):

As shown in Figs. 3 (revised in Fig. 2) and S2 there was strong intra- and inter-experimental variability in several environmental parameters, most prominently temperature, insolation, pH and saturation state, but also salinity and nutrient availability. Further, food supply and faunal/floral composition varied during the experiments as discussed below (Section 4.5) and shown in Werner et al. (2016). This natural variability is an intentional part of the benthic mesocosm set-up as it allows to consider the dynamics of benthic communities reacting to environmental changes under near-natural boundary conditions (Wahl et al., 2015, 2016). On the other hand, the lack of control on several environmental parameters also has drawbacks for the interpretation, comparability and reproducibility of results from different seasonal experiments. As described in Section 2.3, we use the term "seasonal factors" to collectively describe variations of experimental conditions between the four experiments, including environmental parameters and the ontogenetic development of S. spirorbis. While some of these factors are clearly dominated by seasonal change (e.g. light, temperature), others may vary on different time scales. Without multi-annual replicates we can not prove the seasonal nature of the observed changes in S. spirorbis growth between the four experiments. We therefore use the term "seasonal" as a simplifying descriptor of inter-experimental changes, although their seasonal nature needs to be verified in future multi-annual experiments.

*Reviewer specific comment 1*: Reproduction occurs between spring an summer. Therefore the authors are comparing not just juveniles and adults, but different generations that can have different sensitivities due to different starting (acclimation) conditions.

*Response S1*: **This is a very good remark. According to published work on Spirorbis and our own results reproduction occurs in several intervals between late spring and autumn. We agree that the different conditions during the initial growth of each generation may cause different acclimation with respect to, e.g., saturation state. This could result in a better pH tolerance of the late summer and autumn generations that started to grow under lower pH conditions than the late Spring generations. Unfortunately, as stated in the comments of VBSC Chan and in Response G3/S2, we have no replicates for the seasonal experiments, as our study covered only one year. Therefore, our data do not allow to draw conclusions about different acclimation at different seasons.**

*Reviewer specific comment 2*: see general comment 3.

*Response S2*: **See response G3/S2 above.**

*Reviewer specific comment 3*: Did the authors check if all specimens were alive at the end of the experimental period?

*Response S3*: **Fucus with attached worm tubes was collected from the basins and freeze dried immediately after the experiments. Complete worms with original (red) colour were visible in many tubes, indicating that they had been alive until freeze drying.**

*Reviewer specific comment 4*: Were alkalinity changes due to increased salinity taken into account? Full carbonate chemistry details should be provided.

*Response S4*: **Alkalinity was measured frequently together with salinity. Alkalinity values of all experiments and treatments were published in Wahl et al. (2015: Fig. 9). In our manuscript alkalinity and salinity are shown in diagrams of the Supplement (Fig. S2). Measured alkalinities were used to calculate the calcite saturation states in the basins (as described in Section 2.5 of our manuscript). We can refer more frequently to Wahl et al. (2015), where appropriate, to better**

**explain the data base for our carbonate chemistry calculations.**

*Reviewer specific comment 5*: Michaelis-Menten type kinetics for the light dependence of diurnal pH variations.

*Response S5*: **We agree that a Michaelis-Menten fit is more appropriate than the simple linear fit. We find a reasonable fit to our data and have modified Figure 4 and the text in Section 3.1 accordingly (see Fig. 3 and Figure caption). In accordance with the previous linear fits, the Michaelis-Menten fit of the high-CO2 treatments shows a half saturation constant that is almost twice the constant of the normal CO2 treatments (1.6 and 0.9, respectively). This indicates a continuing increase of pH amplitudes in the latter under high-light conditions. The corresponding rate constants for normal and high CO2 are very similar, 0.5 and 0.6, respectively.**

*Reviewer specific comment 6*: Specimen at high CO2 and elevated temperature grew twice as much as the specimen from control conditions. This seems counter-intuitive.

*Response S6*: **We agree with the reviewer. However, as shown in Fig. 14, average growth between different winter treatments was not significantly different. As shown in Fig. 12, individual growth rates varied in a wide range in all treatments. So with the high variability observed, growth in the high-CO2 treatments in single specimens can be expected to exceed growth in low-CO2 treatments. Indeed this is illustrated in Figure 9: growth of a single specimen does not represent general (average) growth in a treatment. Note that the specimen shown in (a) had an initial diameter of about 3 mm, while (b) was only about 2 mm wide at the start of the experiment. A shorter tube segment was added to the larger specimen. This is in line with the growth trend shown in Fig. 12. We can add a remark in the figure caption to better describe that high variability of growth: Figure 9. Pristine and corroded S. spirorbis shells.... Note that the specimen in (a) had a larger initial diameter than the specimen in (b), but grew a shorter new**

**tube segment during the experiment.**

*Reviewer specific comment 7*: Units of the y-axis missing in Fig. 14. Juvenile specimens in Fig. 14 are equivalent to "autumn small population". This should be noted in the caption.

*Response S7*: **The y-axis shows growth divided by initial diameter, which is dimensionless (mm/mm). We will add the units (mm/mm) in the diagram (Fig. 4). The figure shows the whole autumn population (small and big). This size difference was only significant in the "small" sub-population (not shown). It was insignificant in the "big" sub-population. However, because the autumn population was dominated by the small sub-population the effect is still significant in the total population. We clarified the figure caption (see below).**

**Revised Figure Captions**

**Figure 1**: A. Two subunits of the Kiel Outdoor Benthocosm with open hood. Subunits with closed hoods are visible in the background on the right. B. Spirorbis spirorbis specimens attached to living brown alga Fucus vesiculosus. Juvenile (white dots, yellow arrows) and adult (white spires, red arrows) specimens of S. spirorbis are visible.

**Figure 2**: Average water temperature, daily insolation, pH and saturation state with respect to aragonite (as proxy for S. spirorbis Mg-calcite) in the four different treatments. Each of the four seasonal experiments is divided into four sub-periods lasting 17-19 days (start and end dates indicated at x-axis). Error bars indicate minimum and maximum values of the mean diurnal cycle during the sub-periods, except for insolation where they indicate day-to-day variability (standard deviation). Insolation was measured at the GEOMAR meteorological observatory (www.geomar.de/service/wetter), about 100 m from the benthocosms.

**Figure 3**: Light dependence of diurnal pH cycles. Average diurnal pH amplitudes in the benthocosm basins for $CO_2$-enriched (left) and ambient (right) treatments plotted

versus the average daily insolation (as in Fig. 3) for the sub-periods of the four seasonal experiments. Dotted lines are Michaelis-Menten fits to the data, y=A*x/(B+x), with rate constants (A) of 0.5 and 0.6 and half saturation constants (B) of 0.9 and 1.6, for ambient and CO2-enriched treatments, respectively.

**Figure 4**: Average growth (Gr/Di) in different treatments during seasonal experiments. In autumn, growth differed significantly between the +T and control treatments. The effect is only significant in the "small" sub-population, while the "big" sub-population showed no significant temperature effect. However, the "small" sub-population dominates the autumn population. Thus the total population shows a significant temperature effect. In summer, no tubes were recovered from the elevated temperature treatments. Results from three-way ANOVA and Tukey's HSD tests; * significant difference (p <0.05).

―――――――――――――――――――――――

[Figure]

[Figure]

Figure 1

**Fig. 1.**

[Figure]

Figure 3

**Fig. 2.**

[Figure]

Figure 4

**Fig. 3.**

[Figure]

Figure 14

**Fig. 4.**

---

## Author Comment (AC2) · 23 Dec 2017

We thank the reviewer for her supportive comments and the constructive review of our manuscript. Below we respond to the reviewer's specific comments.

**Reviewer general comment 1:** This work suggested a seasonal trend of calcification related to maximum irradiance and the respective response to ocean acidification. Unfortunately, these seasonal observations were not replicated and should be interpreted with caution.

**Response G1**: It is correct that we suggested, based on our data, a seasonal trend in Spirorbis growth rates. It is well visible in our Fig. 14 with highest average growth rates in the summer and autumn experiments declining to lowest growth in the spring experiment. However, we do not suggest that this trend is related to irradiance or acidification. If this were the case we would expect the highest growth rates in experiments with highest pH/saturation state. On the contrary, we observe high growth rates in autumn coinciding with very low irradiance and lowest pH (see attached Figure 1, i.e. revised Fig. 3 of the manuscript). In fact, the lowest mean growth rate was observed in spring when saturation state was the highest of all experiments and irradiance was intermediate (Figure 1). In addition, our experiments with elevated pCO2 show no significant influence of acidification on growth rates.

Further, as shown in Fig. 12 of the manuscript, growth strongly depends on the size of the worm tubes, i.e. on the ontogenetic status of the worm. Small (juvenile) worms show highest growth rates. Rates generally decrease with increasing age and size. The dominance of juveniles in summer and autumn is responsible for the high growth rates in these experiments, despite a very low saturation state and low light conditions in autumn.

Our conclusion with respect to growth rates therefore is that growth patterns are dominantly controlled by ontogenesis (genetically controlled) and only modified by external parameters. We nevertheless agree that the lack of multi-annual data and seasonal replicates in our experiments limits our interpretations concerning seasonal impacts on Spirorbis growth.

To more clearly point out these limitations we will change the title of Section 3.5.2 to "Differences between seasonal experiments" and add the following text to Section 4:

As shown in Figs. 3 and S2 there was strong intra- and inter-experimental variability in several environmental parameters, most prominently temperature, insolation, pH and saturation state, but also salinity and nutrient availability. Further, food supply and faunal/floral composition varied during the experiments as discussed below (Section 4.5) and shown in Werner et al. (2016). This natural variability is an intentional part of the benthic mesocosm set-up as it allows to consider the dynamics of benthic communities reacting to environmental changes under near-natural boundary conditions (Wahl et al., 2015, 2016). On the other hand, the lack of control on several environmental parameters also has drawbacks for the interpretation, comparability and reproducibility of results from different seasonal experiments. As described in Section 2.3, we use the term "seasonal factors" to collectively describe variations of experimental conditions between the four experiments, including environmental parameters and the ontogenetic development of S. spirorbis. While some of these factors are clearly dominated by seasonal change (e.g. light, temperature), others may vary on different time scales. Without multi-annual replicates we can not prove the seasonal nature of the observed changes in S. spirorbis growth between the four experiments. We therefore use the term "seasonal" as a simplifying descriptor of inter-experimental changes, although their seasonal nature needs to be verified in future multi-annual experiments.

**Reviewer major comment 1:** The current statistical approach is prone to misleading conclusions on the effects of season and treatments.
- The lack of sound independently replicated seasonal experiments limits possible conclusions.
- Approach in statistical analysis may overestimate the treatment effects from oversampling

pseudo-replicates.
The authors should carefully modify their interpretations and make sure that their observations are supported by sound statistics.

**Response M1**: As stated above, we agree that without independently replicated seasonal experiments we cannot conclude that differences observed between our 4 experiments (spring, summer, autumn, winter) have a truly seasonal nature. This means that we can only show that differences between experiments are statistically significant, but we cannot say whether the same differences would occur in the same way in subsequent years (see Response G1).
On the other hand, treatment effects (effects of changed $pCO_2$ and/or temperature) were not compared between seasons. In the statistical analysis (three-way ANOVA) we included "season" as the third factor (in addition to T and $CO_2$) for the analysis, but interpretation and Tukey's HSD tests did not include the factor "season". Comparisons were carried out only between different treatments within a single season, using the median value of three replicated basins. All replicates used for interpretations (three independent basins for each of the four treatment conditions: control, +T, +$CO_2$, +T+$CO_2$) are true, independent replicates. As one exception (Section 3.5.2, last paragraph) we used three-way ANOVA to compare treatment influences on growth of the adult populations in the winter and autumn experiments. However, the results showed only insignificant differences ($p>0.67$).
In all other cases interpretations of differences between the seasonal experiments, as described in Response G1, do not include these treatment effects. Consequently Fig. 14 only indicates the significant difference between treatments within a single season, while the much larger differences between seasonal experiments are not statistically tested. We interpret the latter as dominantly reflecting ontogenetic growth histories.
To better describe the statistical methods applied in our analysis we will modify the last paragraph of Section 2.3 as follows:

Three-, two-, one-way ANOVA and Tukey's HSD tests were used for testing statistical significance of differences between the median values from different treatments and seasons. Each treatment had three replicates but, with a total duration of one year, seasonal experiments were not replicated. Median values were calculated for each of the treatment replicates based on the measured values, resulting in 12 basin medians for every seasonal experiment. In the three-way ANOVA, the three factors were temperature, $pCO_2$ and season. The temperature and $pCO_2$ factors had two levels, elevated and ambient. It should be kept in mind that the season factor here is a multiple factor which includes a range of parameters/conditions such as fjord temperature, pH, saturation state, nutrients and ontogenetic effects of S. spirorbis. Only differences caused by the temperature and $pCO_2$ offsets between the treatments were tested for statistical significance. The "seasonal" factor had no independent (multi-annual) replicates. Differences between seasonal experiments may consequently arise from any of the above mentioned seasonal factors, as well as from other unknown factors.
Assumption of normality of the models' residuals and homogeneity of 10 residual variances were tested with Shapiro-Wilk's tests and box plots, respectively. Statistical analyses were conducted with R (Version 3.2, cran.r-project.org), PAST (Version 3.13, Hammer et al., 2001) and Microsoft Excel (Data Analysis Tool). A probability value of $<0.05$ was considered significant.

**Reviewer major comment 2:** This work has only studied one year to understand potential seasonal impacts, however, authors tried to make a conclusion on seasonal impacts (i.e. Spring, Summer, Autumn, Winter) on the observed responses. This is not appropriate unless there were more than 3 years of observation to make each of spring, summer, autumn, and winter to have 3 independent and random observation. The authors are reminded that any seasonal effect observed is a suggestive trend for future experiments.

**Response M2**: We agree that it would be helpful to investigate seasons in more than a one year campaign. However, since the in-situ mesocosms are dealing with natural fluctuations 3 following years do not really mean that they represent true 'replicates'. They may differ in boundary conditions. In our delta approach (Wahl et al., 2015), by carefully comparing un-biased with manipulated experiments, we obtain information about the impact of changing T, PCO2 and T+PCO2. In addition we observe seasonal and biogeochemically well characterized variations between our experiments (Spring, Summer, Autumn, Winter). However, the 'seasonal effect' we show here includes a range of different factors such as light, food, size, generations, that may also affect the growth rate, but are not included in the experimental settings (no repeats), that's why we actually focus on comparing the treatment parameters within one season, but not between seasons (e.g. Fig. 14).

Nevertheless, as we show in our manuscript, the very strong seasonal cycle of temperature and insolation and its impact on biological systems in the Baltic Sea needs to be considered for interpreting growth conditions of Spirorbis. The reproduction cycles of Spirorbis are well documented in the literature and are known to follow a seasonal timing (see Section 4.2). Therefore, neglecting seasonal information in our data would lead to misinterpretations. Accordingly, seasonal impacts have been described in several publications about different aspects of the same benthocosm experiments, e.g. Raddatz et al. (2017), Werner et al. (2016), Graiff et al. (2015), Al-Janabi et al. (2016, Mar. Biol., 163, 14). As explained in Responses G1 and M1 we will add notes of caution about the significance and interpretation of differences between the seasonal experiments in Sections 2.3 and 4.

**Reviewer major comment 3:** The interpretation of the result is not appropriate. The experiment is designed to appropriately answer whether there were effects by the 4 levels of treatment (ambient, pCO2, +T, and pCO2+T), each benthocosm served as independent replicates for statistical analysis. HOWEVER, samples taken within the samples mesocosm are NOT independent replicates. I suggest the authors take the average value for every parameter they measure as the value in each replicate and perform 2-way ANOVA or appropriate statistical test, take n=3. Over-sampling and pseudoreplication may lead to an overemphasis of the effects of treatment effects.

**Response M3**: We agree that individual worm tubes from a basin are not independent replicates when evaluating treatment effects. Indeed, we used the median value of each basin for statistical analyses (ANOVA), i.e. n=3 for every treatment. This is explained in Section 2.3 (see Response M1 for an improved version of Section 2.3).

**Reviewer major comment 4:** It is important to address a more general theory how calcifiers may benefit from the presence of photosynthetic activities of macroalgae in the other ecosystems.

**Response M4**: In our manuscript we describe the potential impact of pH changes in the algal boundary layer. Other effects of algal activities were beyond the scope of our study.

**Reviewer major comment 5:** The impact of pCO2 on the photosynthetic activity of Fucus is also an important point of discussion. Here are a few studies which are relevant to this topic of discussion.

**Response M5**: The impact of elevated pCO2 on Fucus in the same experiments was studied by our colleagues and is published in Graiff et al. (2015, Frontiers in Marine Science 2, 00112, doi: 10.3389/fmars.2015.00112). We thank the reviewer for the two additional references, especially for the paper of Raddatz et al., which we will add to our manuscript. The increased anti-fouling

activities of Fucus in the elevated temperature summer experiments may have contributed to the high Spirorbis mortality in these treatments (line 3 of Section 4.4).

We agree that discussing pCO2 impacts is important for interpretations concerning the Fucus ecosystem (see Graiff et al., 2015). However, we saw no significant impact of pCO2 of S. spirorbis tube growth. Therefore discussing impacts of elevated pCO2 on Fucus algae and their consequences for Spirorbis growth would be rather speculative.

**Reviewer detailed comment 1:** Figure 5A, 12, 13, and 15, should be removed or plotted again after inappropriate interpretation of seasonal response and pseudoreplication are eliminated.

**Response D1**: We do not agree with the reviewer. All of these figures show important information. Figure 5A shows the size distribution of all analysed Spirorbis tubes from all experiments. It shows that there were systematic size differences between the four experiments. The latter are color coded to show these differences. This is very important for the interpretation of the growth data, because growth strongly depends on tube size, as is shown in Fig. 12. There is no statistical evaluation or significance test in Figure 5A. The differences in size distributions between the seasonal experiments are mainly reflecting the ontogenetic evolution and reproduction cycles of S. spirorbis. As discussed in Section 4.2 the seasonal and episodic nature of Spirorbis reproduction is well documented in the literature, which justifies our interpretation.

Figure 12 shows a general relationship between the initial worm tube diameter and tube growth. We conclude from this relationship that growth is primarily limited by ontogenesis. The dependence is valid for all specimens of all our experiments, regardless of season or treatments. Therefore using n=2783 for the linear fit in Fig. 12 is correct. As in Figure 5 we added color coding and average values to show that initial size distributions and growth differed systematically between the different experiments (seasons). This is very important for the data interpretation. In our opinion it primarily reflects the seasonally controlled life cycle (reproduction and ontogenesis) of S. spirorbis. We do not draw conclusions about impacts of acidification or warming from them (see also Response G1).

Figure 13 shows a dependence of shell weight on saturation state. There was a significant positive correlation in the spring and autumn experiments. However, it also shows that an additional factor had a strong impact on weight increase. Weight increase was similar in the spring and autumn experiments although in spring the water was oversaturated while in autumn it was undersaturated with respect to Spirorbis Mg-calcite. Very likely this additional factor is again related to the ontogenetic development of the worms.

Figure 15 compares growth in juvenile populations from different experiments (summer and autumn). As stated in Response G1 we can not prove that the observed differences are truly seasonal and would be similar between summer and autumn if repeated in subsequent years. However, the differences between the two compared experiments are statistically significant and each experiment had three independent replicates (control treatments, 3 basins).

**Reviewer detailed comment 2:** Page 6 Line 2 This selection of subpopulation needs to be further justified.

**Response D2**: Growth of Spirorbis tubes depends strongly on the initial diameter (see Figure 12). Therefore, it is essential to compare populations with similar initial diameters if differences in growth in different treatments are to be detected. We therefore analysed only a subset of data from the two experiments, for which the initial diameters ($D_i$) were in a similar range. We will change the sentence on page 6, line 2-3:

In order to derive comparable populations with similar $D_i$ in the summer and autumn-small data

we selected sub-populations that had similar D_i ranges and similar median D_i values. Tubes outside this D_i range were not used in the statistical analysis.

**Reviewer detailed comment 3:** Page 12 Line 5-9 how did the researcher ensure the growth measurement of broken and strongly damaged tubes was accurate?

**Response D3**: Broken and strongly damaged specimens from the summer experiments with elevated temperatures were not analysed (e.g., see Fig. 14, Table 2). In other experiments shells were mostly intact. Broken or damaged tubes were not measured (e.g., indicated by "no data" in Figure 14).

**Reviewer detailed comment 4:** Page 4 Line 23 How to quantify the volume of the blades, what has been done to ensure they are similar volume? How many worms were present in each replicate tank?

**Response D4**: Individual Fucus plants were selected by visual inspection to contain approximately the same volume of blades and similar amounts of Spirorbis. That means that plants of similar sizes and with similar numbers of blades were collected. However, it is not possible to collect 12 identical Fucus plants in a natural environment.
Typical densities of Spirorbis tubes at the start of the experiments is shown in Fig. 1. Average starting populations were on the order of 100-200 specimens per tank.
We will add this information to Section 2.1.

**Reviewer detailed comment 5:** Page 4 Line 26-27 provide the size/ volume of the boxes and duration of staining, at what density were the animal kept in the staining solution? Was the pH value before and after the staining monitored? Please provide this information for repeatability.

**Response D5**: The twelve Fucus thalli with attached Spirorbis collected for each experiment were stained in a 10 L transparent plastic box for three days. We did not measure pH during staining, as the box was continuously bubbled with ambient air to keep pH stable. The pH of the seawater used for staining varied from about 7.7 in autumn to 8.5 in spring. We added 0.3 L of calcein solution adjusted to a pH of 8.1, thereby altering the seawater insignificantly by less than 0.01 pH units.
We will add the box volume in Section 2.2.

**Reviewer detailed comment 6:** Page 6 Line 2 provide statistics for the difference

**Response D6**: Will be added: ...medians differed significantly (two-way ANOVA, p=0.019).

**Reviewer detailed comment 7:** Page 6 Line 16 list polishing materials and duration for reproducibility

**Response D7**: We will add to Section 2.4:
Samples were wet polished with grinding paper followed by polishing solutions of 9 μm, 3 μm and 1 μm grain size until no more scratches were visible on the polished surface.

**Reviewer detailed comment 8:** Page 6 Line 21 please provide excitation and emission wavelength, and resolution of images acquired

**Response D8**: Images were recorded with a resolution of 1360*1024 pixels. Excitation wavelength was 495 nm. Emission (517 nm) from calcein was recorded.
This information will be added in Section 2.4.

**Reviewer detailed comment 9:** Page 6 Line 27 pH calibration was at the lower range (pH 4 and pH 6.865), this requires extrapolation when measuring the ocean pH at 7.4-8.0. it is recommended to check the accuracy of this approach with another meter that has been calibrated by 3 points (for example pH 4, 7 10)

**Response D9**: The field measurements were conducted in this study with 2 NBS calibration solutions kept at in-situ temperature. The pH 9/10 buffer was avoided to prevent impact of possible $CO_2$ contamination under field conditions. The stability of the electrodes` Nernst slope and the applicability of the 2-point calibration to higher pH was previously tested by the measurement of a pH 9 calibration solution. Independently calculated pH values (with CO2sys) based on measured dissolved inorganic carbon and total alkalinity values showed good agreement with the measured pH in the range between 8 to 9 (Wahl et al., 2015; Limnol. Oceanogr. Meth.).
We will add this to Section 2.5.

**Reviewer detailed comment 10:** Page 6 Line 31, what is the importance of filtering and adding NaCl to TA samples? Please supplement the reason.

**Response D10**:
Filtering is necessary to remove microbes and particles that could alter the sample TA. Filtering does not alter the TA value. NaCl was added to adjust the ionic strength of the acid used in the titration procedure to avoid changes in ionic strength during analysis. Information will be added to Section 2.5.

**Reviewer detailed comment 11:**. Page 7 Line 1 should it be "Both DIC and TA measurements are calibrated. . ."?

**Response D11**: Yes, standards were used for both parameters. We will add: ... for DIC and TA.

**Reviewer detailed comment 12:** Page 7 Line 13 what was the technique used for measuring Ca Si and P?

**Response D12**: We will add:
The dissolved concentrations of Si, P and Ca were analysed by inductively-coupled plasma optical emission spectrometry (iCAP 6300 DUO, Thermo Fisher Scientific) after appropriate dilution. The accuracy and precision was routinely checked with the certified seawater standard CASS-5 as previously described (Kowalski et al., 2012, Est. Coast. Shelf Sci 100). PO4 was also measured by spectrophotometry using a QuAAtro nutrient analyser (SEAL Analytical; Winde et al. 2014; J Mar Sys). Accuracy and precision checked by replicate analyses of a solution from powdered phosphate salts were better than 8% RSD.

**Reviewer detailed comment 13:** Page 9 Line 6-21 The use of adult and juvenile is not appropriate if the literature shows that sexual maturity begins at 1.9 m. Authors are recommended to use neutral words such as small and large cohort/ young and old cohort

**Response D13**: We agree that using the term "adult" is problematic, as the selected size range may include juvenile specimens. We will replace the terms by "small" and "large", where appropriate.

**Reviewer detailed comment 14:** Page 9 Line 22-29 Please explain the use of "mode" instead of "mean"

**Response D14**: In this paragraph we describe the most common sizes of the different Spirorbis populations. This is correctly described by the population mode, not by the mean. The statistical term "mode" describes the most frequently occurring class in a distribution, which can differ significantly from the mean in skewed or non-normal distributions. The modes of the populations can easily be identified in Figure 5.

**Reviewer detailed comment 15:** Page 10 Line 15 Please reword and clarify the meaning of this line

**Response D15**: "The sentence will be changed to: "The newly grown lamellae cover a large area along the inner tube wall surface, while little new material is attached to the outer tube wall surface."

**Reviewer detailed comment 16:** Page 10 Line 21 "Tube opening" not "tube mouth"

**Response D16**: Will be changed.

**Reviewer detailed comment 17:** Page 10 Line 24 and Figure 10 Please provide sample size that supports this percentage measurements

**Response D17**:[see comment D26 for Table 2]

**Reviewer detailed comment 18:** Page 12 Line 23 to Page 13 Line 6 The seasonal effects is not supported by independent replication.

**Response D18**: We agree with the reviewer that without independently replicated seasonal experiments we cannot conclude that differences observed between our 4 experiments (spring, summer, autumn, winter) have a truly seasonal nature (see Response G1, M1, M2). As noted above we will change the title of Section 3.5.2 to "Differences between seasonal experiments" and add text to more clearly point out these limitations.
As discussed in Section 4.2 of our manuscript, the seasonal nature of Spirorbis reproduction is well-documented in the literature. So our interpretation of the observed ontogenetic differences as being seasonal is justified.

**Reviewer detailed comment 19:** Figure 1 please add arrows to show juvenile and adults
**Reviewer detailed comment 20:** Figure 2 Please make the font bigger

**Reviewer detailed comment 21:** Figure 7 Please improve the font choice for "Growth direction" and " Tube wall inner surface", it is not easy to read
**Reviewer detailed comment 22:** Figure 8 The labels for the longitudinal section and the cross section are mixed up, note that longitudinal section is the cut made along the long axis of the tube.
**Reviewer detailed comment 23:** Figure 9b - please added annotation to indicate the microboring structures as mentioned in Page 11 Line 12

**Response D19-23**: Figures will be changed.

**Reviewer detailed comment 24:** Figure 12 use of red triangle and red circle makes the pattern hard to see, n=2783 is an example of pseudoreplication, please re-examine the result and make a new plot

**Response D24**: Please note that n=2783 is not the number of replicated treatments but the number of specimens used in our data set. The trend line describes the general dependence of growth on initial diameter, which is independent of the treatments in our experiments. Our interpretation for this trend is the ontogenetically controlled growth behaviour of Spirorbis, which is of course in sync with the seasons, but NOT with calcite saturation or pH.
We will change the colors of the symbols. The autumn-big triangles were supposed to be brown, but rather look red in the PDF.

**Reviewer detailed comment 25:** Figure 16 fonts for the scale bar are too small

**Response D25**: Figure will be changed.

**Reviewer detailed comment 26:** Table 2 - was is the total number of worms sampled for this percentage? Please provide the numbers by expanding the table.

**Response D26**: Expanded Table 2 is attached.

**Typos:**

Page 4 Line 9 "In this area," comma is missing
Page 5 Line 11 "In total,"
Page 7 Line 33 it is clearer to say " with respect to S. spirorbis tube Mg-calcite"

**Response**: Thank you! Typos will be corrected.